# Spatial molecular-dynamically ordered NMR spectroscopy of intact bodies and heterogeneous systems

Kengo Ito [1,2], Yuuri Tsuboi[1] & Jun Kikuchi [1,2,3 ✉]

Noninvasive evaluation of the spatial distribution of chemical composition and diffusion behavior of materials is becoming possible by advanced nuclear magnetic resonance (NMR) pulse sequence editing. However, there is room for improvement in the spectral resolution and analytical method for application to heterogeneous samples. Here, we develop applications for comprehensively evaluating compounds and their dynamics in intact bodies and heterogeneous systems from NMR data, including spatial $z$-position, chemical shift, and diffusion or relaxation. This experiment is collectively named spatial molecular-dynamically ordered spectroscopy (SMOOSY). Pseudo-three-dimensional (3D) SMOOSY spectra of an intact shrimp and two heterogeneous systems are recorded to evaluate this methodology. Information about dynamics is mapped onto two-dimensional (2D) chemical shift imaging spectra using a pseudo-spectral imaging method with a processing tool named SMOOSY processor. Pseudo-2D SMOOSY spectral images can non-invasively assess the different dynamics of the compounds at each spatial $z$-position of the shrimp's body and two heterogeneous systems.

[1] RIKEN Center for Sustainable Resource Science, 1-7-22 Suehiro-cho, Tsurumi-ku, Yokohama, Kanagawa 230-0045, Japan. [2] Graduate School of Medical Life Science, Yokohama City University, 1-7-29 Suehiro-cho, Tsurumi-ku, Yokohama, Kanagawa 230-0045, Japan. [3] Graduate School of Bioagricultural Sciences, Nagoya University, 1 Furo-cho, Chikusa-ku, Nagoya, Aichi 464-0810, Japan. ✉email: jun.kikuchi@riken.jp

Compared with other analytical methods, nuclear magnetic resonance (NMR), magnetic resonance imaging (MRI), and magnetic resonance spectroscopy (MRS) have a very broad range of uses. Solution-state NMR is especially useful for the determination of chemical structure and for composition analyses[1–3]. As well as its importance for synthetic and natural product chemistry, comprehensive compositional analysis plays a significant role in the field of the metabolomics, where it is used to evaluate the characteristics of complex samples or mixtures and the differences between them[4]. The complex NMR signals of metabolites can be identified through comparisons with database records and computation[5–8]. Insoluble polymers and molecules that do not have an affinity for solvents are often evaluated using solid-state NMR (SS-NMR) based on magic angle spinning (MAS)[9]. MRI and MRS have also made a major contribution in the field of medicine because they can be used for the noninvasive evaluation of spatial $^1H$ density, diffusion or relaxation, and chemical shift in biological and other samples[10,11]. In this background, the Nobel Prize in Chemistry 1991 was awarded to Richard R. Ernst "for his contributions to the development of the methodology of high-resolution nuclear magnetic resonance (NMR) spectroscopy"[12]. This technology has become a very powerful tool in the identification of molecular structures in organic chemistry. It also has been put to practical use in MRI in the medical field and has been a powerful clue for diagnostic imaging. Furthermore, the Nobel Prize in Physiology or Medicine 2003 was awarded to Paul Lauterbur and Peter Mansfield "for their discoveries concerning magnetic resonance imaging"[12]. However, MRS is still a developing technology and is associated with problems such as low resolution and the low sensitivity of signals due to dipole–dipole coupling and inhomogeneity of the magnetic susceptibility in a sample.

High-resolution MAS (HR-MAS) can be used to obtain HR spectra even from intact samples that have not been crushed, a great advantage for noninvasive composition evaluation[13,14]. In recent years, studies have reported the HR evaluation of spatial composition distribution of insect samples using chemical shift imaging (CSI) with HR-MAS[15,16]. As well as chemical shift information, NMR can also be used to evaluate aspects of the physical properties of compounds, such as their diffusion and magnetic relaxation. Diffusion-ordered spectroscopy (DOSY) is a pseudo-two-dimensional (2D) NMR technique that diffusion coefficients can be extracted from pulsed-field gradient-attenuation profile collected in DOSY experiments. This technique is useful for mixture analysis because it allows the separation of signals of compounds with different molecular weights[17]. Relaxation-ordered spectroscopy (ROSY) such as saturation recovery (SR), inversion recovery (IR), Carr–Purcell–Meiboom–Gill (CPMG), periodic refocusing of $J$ evolution by coherence transfer (PROJECT), and relaxation-encoded selective total correlation spectroscopy (REST) are pseudo-2D NMR techniques that evaluate spin–spin relaxation ($T_2$) and spin–lattice relaxation ($T_1$), making them useful for signal separation and the assignment of compounds in a mixture[18].

Recently, a new NMR measurement method has been developed that combines CSI and DOSY. This method allowed the comprehensive evaluation of the spatial characteristics, composition, and diffusion of a lithium ion battery[19]. However, the resolution and analysis of this method have room for improvement, and as yet there has been no report of the method being applied for the comprehensive evaluation of a biological sample, in which the diffusion or relaxation of each compound in the space was analyzed with a HR spectrum. Also, as the similar study, diffusion-weighted magnetic resonance imaging (DW-MRSI) was known as a unique tool for the noninvasive exploration of the structure and physiology of the intracellular space in vivo. DW-MRSI is the method for measuring and quantifying the diffusion properties of compounds in the intracellular space, contributing to the understanding of DWI. Information can be provided to help characterize the various pathological mechanisms of the disease. However, few implementations of in vivo DW-MRSI have been published yet[20].

This report proposes new pulse sequences collectively named spatial molecular-dynamically ordered spectroscopy (SMOOSY) for a technique that is driven by diffusion, relaxation, and REST encoding. The CSI and SMOOSY used in this study has only information of the spatial $z$-position from one-dimensional (1D) imaging and does not include the $x$- and $y$-positions from three-dimensional (3D) imaging like the MRI. The pseudo-spectral imaging method as a new tool named the SMOOSY processor for spectrum processing and analysis is also proposed. As examples of possible applications, we describe comprehensive analyses of an intact biological sample and two heterogeneous systems such as the diffusion of feeding stimulants from aquaculture feeds and transport of small molecules via a membrane filter. We also present an example of spectrum analysis using the SMOOSY processor.

## Results

**Overview of SMOOSY experiments**. The method referred to here as SMOOSY, was driven by diffusion encoding ($D$-SMOOSY as well as CSI-DOSY), $T_1$ encoding ($T_1$-SMOOSY as well as CSI-SR), $T_2$ encoding ($T_2$-SMOOSY as well as CSI-PROJECT), $REST_1$ encoding ($REST_1$-SMOOSY as well as CSI-$REST_1$), or $REST_2$ encoding ($REST_2$-SMOOSY as well as CSI-$REST_2$). The basis of the pulse sequence is shown in Fig. 1a. A magnetic field gradient pulse of length δ and strength $k$ in the spin echo τ encodes the signal with respect to the spatial $z$-position[21]. Applying DOSY, SR, PROJECT, $REST_1$, or $REST_2$ in front of this CSI pulse sequence (Fig. 1b–e) allows diffusion and magnetic relaxation phenomena in spatial $z$-position to be analyzed. In other words, SMOOSY can be thought of as of DOSY and ROSY with added spatial z-position information (Fig. 1f). The biological sample and two heterogeneous systems contained a large amount of water, so the water signal was suppressed by a pre-saturation pulse. When an intact biological sample is used, there is notable signal broadening due to dipole interactions and the inhomogeneity of the magnetic susceptibility; using HR-MAS suppresses this effect. More details of pulse sequences are shown in Supplementary Fig. 1 and Supplementary Table 1. The pseudo-3D spectra obtained by the SMOOSY experiments were converted to the pseudo-2D SMOOSY spectral image by performing exponential curve fitting[22,23] of diffusion or relaxation dimension and the field of view (FOV)[24] calculation of spatial $z$-position using the developed SMOOSY processor (Supplementary Figs. 2 and 3).

**SMOOSY experiments using homogeneous samples**. Several concerns of the SMOOSY experiments were investigated for applying these experiments to an intact biological sample. For the pulse sequence of $D$-SMOOSY, we were concerned whether the imaging gradients in CSI affected the diffusion measurements and diffusion coefficient calculation. However, the traditional pseudo-2D DOSY spectrum and sliced pseudo-2D DOSY spectrum from the pseudo-3D $D$-SMOOSY spectrum using a homogeneous sample produced the same results including diffusion coefficients; imaging gradients in CSI did not affect diffusion measurements or calculation of the diffusion coefficient (Supplementary Fig. 4). The effect of different MAS on results was evaluated in DOSY experiments using homogeneous sample (Supplementary Fig. 5). In the DOSY experiment, a comparison of the pseudo-2D DOSY

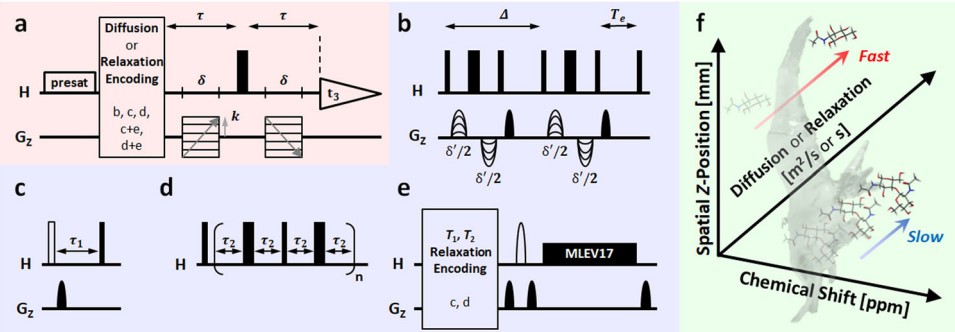

**Fig. 1 Pulse sequence of SMOOSY used in this study. a** SMOOSY was driven by **b** diffusion encoding (*D*-SMOOSY as well as CSI-DOSY), **c** $T_1$ encoding (*$T_1$*-SMOOSY as well as CSI-SR), **d** $T_2$ encoding (*$T_2$*-SMOOSY as well as CSI-PROJECT), or **e** REST encoding (REST-SMOOSY as well as CSI-REST). REST was also driven by $T_1$ or $T_2$ encoding. **f** This method provides information, including spatial *z*-position from 1D-imaging, chemical shift, and diffusion or relaxation. Further details are given in Supplementary Fig. 1 and Supplementary Table 1.

spectrum measured at 3 kHz with the MAS frequency and the pseudo-2D DOSY spectrum measured at 6 kHz with the MAS frequency showing the change in the diffusion coefficient. The diffusion coefficient when the MAS frequency was 6 kHz was approximately $3.08 \times 10^{-10}$ (m2/s) larger than the diffusion coefficient when the MAS frequency was 3 kHz. From this result, a rapid MAS frequency affected the diffusion coefficient of compounds to be fast. In SS-NMR, the rate of MAS changes the $T_1$ and $T_2$ relaxation times[25,26]. The relationship between MAS and diffusion coefficient or relaxation time was considered to be proportional. Therefore, although the diffusion coefficient and relaxation time changed depending on MAS, it was considered that the evaluation of the relative profile was also possible in DOSY, ROSY, and SMOOSY experiments.

**Evaluation of diffusion and relaxation of compounds at each spatial *z*-position in the shrimp's body.** In this study, an intact shrimp (*Palaemon sp.*) was used as the biological sample. It was approximately 1.2 cm long and up to 4 mm wide, making it suitable for inclusion in the 4 mm HR-MAS rotor. Deuterated methanol (MeOD) solvent was used to facilitate locking for the NMR measurement and to prevent the elution of components by enzymes in the shrimp with tissue fixed in methanol. The inner lid was inserted into the rotor to fix the spatial *z*-position of the biological sample and to prevent MeOD solvent from leaking. A MAS frequency of 3 kHz was considered suitable[14]. At the time of the measurement, the temperature was 299 K. Details of the experimental conditions are described in the experimental section.

At first, ¹H–¹³C heteronuclear single quantum coherence (HSQC) spectroscopy was used to examine the major compounds contained in the intact shrimp. Annotation of the HSQC signal was performed using the SpinAssign tool in InterSpin[8] along with the results of previous studies[14,27]. This resulted in the annotation of 114 HSQC signals (Supplementary Fig. 6a and Supplementary Table 2). Furthermore, ¹H–¹³C HSQC-total correlation spectroscopy (HSQC-TOCSY) was used to verify the validity of the annotated signals by chain assignment using TOCSY correlation signals. The usefulness of the chain assignment for complex mixture was reported in our previous study[28]. TOCSY correlation signals of almost all the annotated compounds were confirmed (Supplementary Fig. 7), and the assignment of signals of compounds was considered to be reliable. Among the lipids, the signals for unsaturated fatty acids such as docosahexaenoic acid (DHA) and eicosapentaenoic acid (EPA) were prominent; these are produced by marine microorganisms and concentrated in the shrimp's food chain. In addition, many cholesterol signals

were confirmed, and signals of glycine and proline amino acids were also prominent. Taurine was also observed, consistent with previous reports of taurine in the free state in various animal and plant tissues[29]. Betaine and trimethylamine N-oxide (TMAO), which were also present, are thought to be used by organisms to adjust osmotic pressure[30]. The spectrum of the intact sample using HR-MAS was possible to comprehensively investigate the characteristics of compounds in the sample compared with the spectrum of the extract because it does not depend on the affinity for the solvent.

CSI analyses were performed to determine the major compounds in the intact shrimp. ¹H densities, obtained from the signal intensity values, differed greatly between the head and the tail of the shrimp. Similar results have been obtained in previous studies using MRI[31,32]. There were differences and similarities in the spatial *z*-position distributions of the various components detected. (Supplementary Fig. 6b). Cholesterol, fatty acids, taurine, betaine, and TMAO were observed along the entire spatial *z*-position. The signal strengths and half-widths of unsaturated fatty acids such as DHA and EPA were greater toward the head. triacylglycerol (TAG) was observed almost exclusively toward the head, and phosphatidylcholine was concentrated mainly toward the tail.

The diffusion or relaxation of the components in the spatial *z*-position was evaluated by the pseudo-2D SMOOSY spectral image produced using SMOOSY processor. *D*-SMOOSY was able to capture differences in the diffusion phenomena of each compound at different spatial *z*-positions (Fig. 2). Lipids were present along the entire spatial *z*-position, but the diffusion coefficient was lower toward the head than toward the tail ($1.58 \times 10^{-10}$ (m2/s) vs. $1.00 \times 10^{-9}$ (m2/s), respectively). This may have been due to the influence on diffusion of the interaction between lipids and proteins. In addition, the motility of lipid changes inside and outside the cell membrane, which may reflect its existence state. Or it may reflect the state of the organ. Furthermore, the difference in the diffusion of these compounds at each spatial *z*-position of the intact shrimp is thought to be due to the volume effect and the amount and interaction of chitin constituting the shell. For example, in a paper studying the molecular structure of fungal cell walls by SS-NMR, components closer to the cell membrane, such as chitin and glucan, had lower motility and were affected by relaxation time[33]. In this study, SS-NMR was used to evaluate the chemical shift and relaxation time of chitin, which constitutes the shrimp shell (see Supplementary Fig. 8 and Supplementary Discussion); however, it was impossible to evaluate chitin due to the problem of fast relaxation time in CSI using HR-MAS. If these macromolecules and low molecules can be evaluated by imaging at the same time, this can to lead to

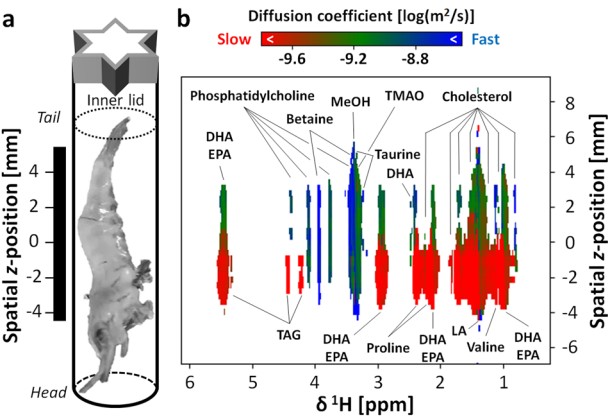

**Fig. 2 The *D*-SMOOSY experiment. a** The sample was an intact shrimp placed in the HR-MAS rotor. The black bar shows the range of detected spatial *z*-position. **b** Pseudo-2D *D*-SMOOSY spectral image. The annotated major compounds are shown with diffusion coefficients colored from red (slow diffusion) to blue (fast diffusion) according to the range shown in the bar above the figure. MeOH denotes methanol as the fixative and solvent.

new discoveries. For TAG, the diffusion coefficient was lowest for the TAG concentrated toward the head. TMAO, taurine, and betaine were present throughout the spatial *z*-position, but there were significant differences in the diffusion coefficients at different spatial *z*-positions. It is possible that these substances have little interaction with other components and so these results may reflect their condition in the free state. However, methanol as a fixative and solvent appeared at 3.37 ppm. The signals of compounds appeared nearby here may have affected the diffusion coefficient by signal overlap. Phosphatidylcholine, which is concentrated toward the tail, had a large diffusion coefficient of approximately $2.51 \times 10^{-9}$ (m²/s). The signal intensity of glycine in intact shrimp was reduced rapidly upon gradient strength dimension. Optimization of the exponential curve fitting for the signal of glycine did not converge and failed, and this signal image on the *D*-SMOOSY spectral image is not shown. The reason for this was considered to be that glycine was so quick to diffuse or may have a special motility because this compound is used for osmotic pressure adjustment. If experimental NMR parameters are set for target to compound with fast diffusion, this problem may be solved. Therefore, the other pseudo-2D SMOOSY spectral image was also evaluated. This showed differences in the relaxation times of each compound in the spatial *z*-position (Supplementary Fig. 9). With $T_2$-SMOOSY, the lipid profile was similar to that of *D*-SMOOSY, with a shorter relaxation time at the head and a longer relaxation time at the tail. TAG signals with small diffusion coefficients and short relaxation times could not be captured with sufficient quality. Thus, the exponential curve fitting was not well performed in relaxation decay due to fast signal decay. In contrast, glycine, which had a large diffusion coefficient and could not be captured by *D*-SMOOSY, could be evaluated with both $T_2$-SMOOSY and $T_1$-SMOOSY. REST allows the extraction of component sub-spectra from mixtures (see Supplementary Fig. 10 and Supplementary Discussion). Therefore, REST-SMOOSY was considered to also allow the extraction of component sub-spectra from mixtures at each spatial *z*-position.

In this study, the REST-SMOOSY experiment focused on shrimp DHA and EPA. As a result, the relaxation time was shorter at the head and longer at the tail. REST-SMOOSY was considered useful for evaluating individual compounds while avoiding signal overlap. $T_1$-SMOOSY and REST$_1$-SMOOSY were not as pronounced as other SMOOSY techniques used in this

study; however, this experiment may be useful for other samples. The difference between the top and bottom shrimp in CSI and *D*-SMOOSY profiles as well as no noticeable difference in other SMOOSY profiles was considered to be due to the dynamic range problem. Similar to a recent report, in the CSI experiment using intact wasp by MAS at a frequency of 4 kHz, the profile was different between tail, mid, and head. By increasing the echo time (TE), signals with fast relaxation times disappeared, and buried signals appeared[16]. In this study, TE was unified to 1 ms to obtain an overall profile. However, a more different profile was expected to be observed by changing the length of TE.

The damage of the shrimp body by MAS process was evaluated. NMR experiments with MAS from 2 kHz to 6 kHz showed no body damage and no outflow of contents (Supplementary Fig. 11). Similar to a recent report for living organism, MAS at 2.5 kHz for 1 h is the highest speed that could be used without affecting survival[14]. Similar to another report, in the CSI experiment using intact wasp, the wasp body structure was found to be intact after enduring a 4-h span of rotation at 4 kHz[16]. However, it should be noted that faster MAS and using other sample types may have caused damage. In addition, as a result of comparing the pseudo-2D *D*-SMOOSY spectral image measured by these different MAS, although the effect of individual differences was somewhat apparent, the pattern of signal appearance and imaging profile in spatial *z*-position was almost the same in experiments using different MAS. It was confirmed that the diffusion coefficient was slightly changed by different MAS, as mentioned in the experiment with the homogeneous sample. For example, the difference of diffusion coefficient of lipid when the MAS frequencies were 2 and 6 kHz was approximately $9.54 \times 10^{-10}$ (m²/s). There is also concern that MAS may disrupt the internal compounds profile. To evaluate this possibility, a non-rotating HR-MAS experiment was attempted, but analysis was difficult because most of the signals were broad. As a solution to this, there was a recently reported pulse sequence of a HR NMR experiment using a heterogeneous sample in a non-rotating state[34–36]. In addition, the tissue of the biological sample is immobilized, and the measurements will be different from those for metabolic profiling of living tissues. In a recent report on CSI using HR-MAS, a slow sample rotation (500 Hz) experiment acquired metabolic profiling of a living organism in about an hour[37]. Applying such pulse sequences and experimental approaches to SMOOSY may address concerns with the effects of MAS and are considered as areas for future study[37]. In the last section, an application example of the SMOOSY experiment of different samples by NMR in a non-rotating state without using HR-MAS is shown.

**Feature extraction from pseudo-SMOOSY spectra**. Principal component analysis (PCA) of the pseudo-2D *D*-SMOOSY spectral image was performed to characterize differences and similarities in the diffusion phenomena of the compounds in the spatial *z*-position (Fig. 3). The scores on PC1 axis showed the big difference among head and tail (Fig. 3a). From the loadings on PC1 axis, this difference is particularly affected by the diffusion coefficient of lipids such as DHA and EPA, and the diffusion coefficient is large at tail (Fig. 3b). The components that showed little difference in diffusion coefficients were characterized spatially on PC2 axis of score and loading plots. With $T_1$-SMOOSY, $T_2$-SMOOSY, REST$_1$-SMOOSY, and REST$_2$-SMOOSY, differences could not be clearly seen from the spectrum; however, PCA showed it was able to spatially characterize the relaxation phenomena of each compound (Supplementary Figs. 12 and 13). In addition, three-way parallel factor analysis (PARAFAC) of pseudo-3D SMOOSY spectra was also performed and compared

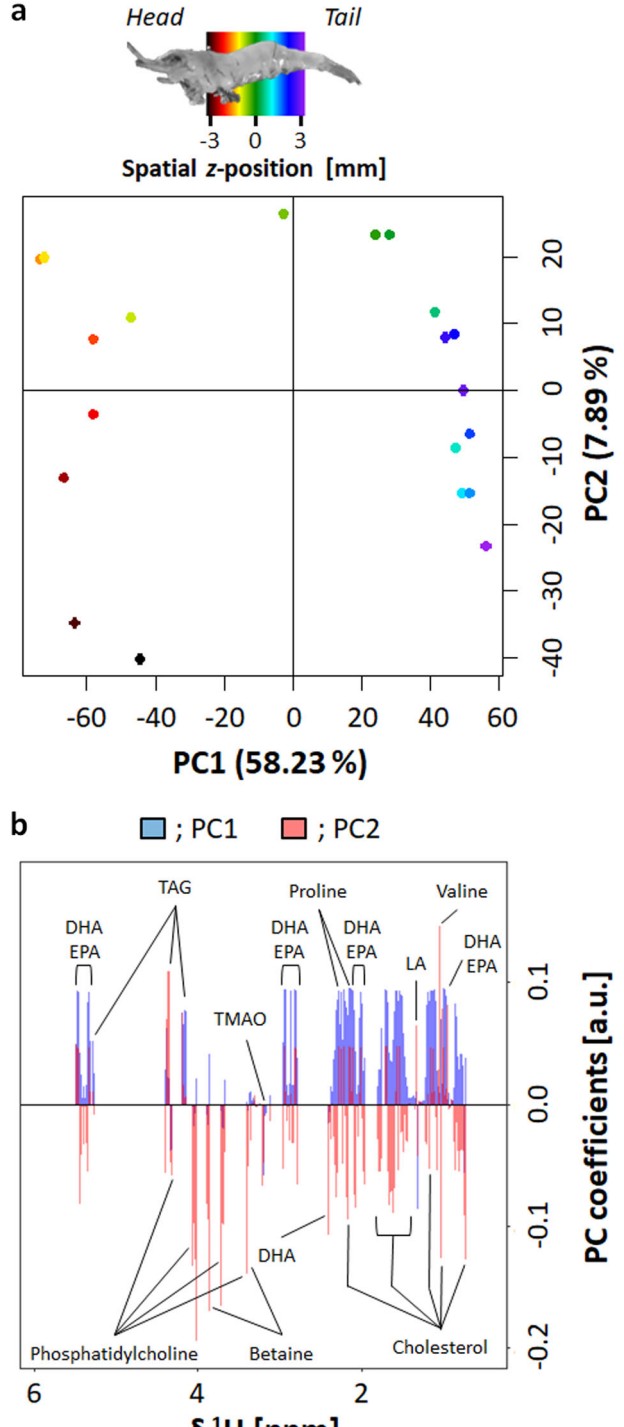

**Fig. 3 PCA using the pseudo-2D *D*-SMOOSY spectral image. a** The scores, colored from black (head end) to purple (tail end). **b** The loadings, colored blue for PC1 and red for PC2.

with the results of PCA using pseudo-2D SMOOSY spectral images. The features of intact shrimp at each spatial *z*-position could be extracted, and the contributions of component 1 and 2 on PARAFAC scores were signal intensities of mainly compounds (Supplementary Figs. 14–16). Thus, PCA using the pseudo-2D SMOOSY spectral image was more suitable for feature extraction of diffusion or relaxation of compounds at each spatial *z*-position than PARAFAC using pseudo-3D SMOOSY spectra.

**SMOOSY experiments by solution-state NMR**. To investigate the further applicability of the SMOOSY experiment, two different samples were used to evaluate the diffusion of components at the *z*-position by *D*-SMOOSY. These experiments were performed with solution-state NMR, which is more commonly used than HR-MAS. The first was a membrane filtration experiment (Supplementary Fig. 17a). In this experiment, a deuterium oxide ($D_2O$) solution containing polyvinyl alcohol (PVA), alanine, and sucrose was prepared, a polytetrafluoroethylene (PTFE) membrane was fixed in an NMR tube, and components in the solution permeated the membrane. From the pseudo-2D SMOOSY spectrum image, it can be seen that PVA having a large molecular weight and a low diffusion coefficient is accumulated at the top of the membrane and hardly passes through it. Alanine and sucrose have permeated the membrane, and that their diffusion coefficients are different. The low molecular weight PVA decomposed to acetate. This has a fast diffusion coefficient and passes through the membrane. It was possible to evaluate whether or not it permeated through the membrane based on the molecular weight and the diffusion coefficient. For this experiment, it was thought that it may be applied to the performance evaluation of a membrane used for sewage treatment and filtration.

Stimulants in aquaculture feeds, such as betaine, organic acids, amines, amino acids, saccharides, and their mixtures, are important for the development of new aquaculture feeds[38–40]. Therefore, the second example was fish feed diffusion (Supplementary Fig. 17b). Feed pellets were placed at the bottom of an NMR tube, $D_2O$ was added, and the diffusion of components of the feed was evaluated. Peptides have a low diffusion coefficient, e.g., $6.81 \times 10^{-11}$ ($m^2/s$) and remain at the bottom of the NMR tube. Small molecules, such as amino acids, organic acids, and saccharides, have large diffusion coefficients; they dissolve and diffuse widely. Especially, the diffusion coefficients for lactate and acetate were large, ca.$1.74 \times 10^{-9}$ ($m^2/s$). Thus, the diffusion of several compounds into the feed was significant with rapid dissolution. To detect the dissolution of compounds into the feed, high-speed measurements immediately after sample preparation is necessary.

Thereafter, $T_2$-SMOOSY was adjusted to about 5 min per measurement. The measurement was started at an early stage after sample preparation, and the dissolution process of the compounds into the feed was assessed (Fig. 4). The dissolution of lactate and acetate was confirmed with relaxation times of about 1.61 s (Fig. 4b). In addition, the dissolution of TMAO, betaine, taurine, choline, creatine, dimethylamine (DMA), and saccharides was confirmed, whereas the dissolution of amino acids was confirmed 35 min after sample preparation (Fig. 4c). A peptide signal was also detected. This finding might be explained by the prolonged relaxation time that was due to the affinity with water or dissolution to improve mobility. After 65 min of sample preparation (Fig. 4d), dissolution and affinity with water proceeded further. These compounds in solution are likely to attract fish. $T_2$-SMOOSY over a short time frame may have problems with deterioration of spatial resolution and detection sensitivity; however, it is sufficient for the preliminary evaluation of samples. These results indicate that the technique is applicable to research on feed development including exploration of attractant compounds for fishes. Details of the above experiments are present in the Methods section and Supplementary Methods.

## Discussion

This paper proposed six SMOOSY pulse sequences. A pulse sequence combining CSI and DOSY has already been reported, but in this study, a highly resolved spectrum was obtained using HR-MAS, and the pseudo-2D spectral imaging method has

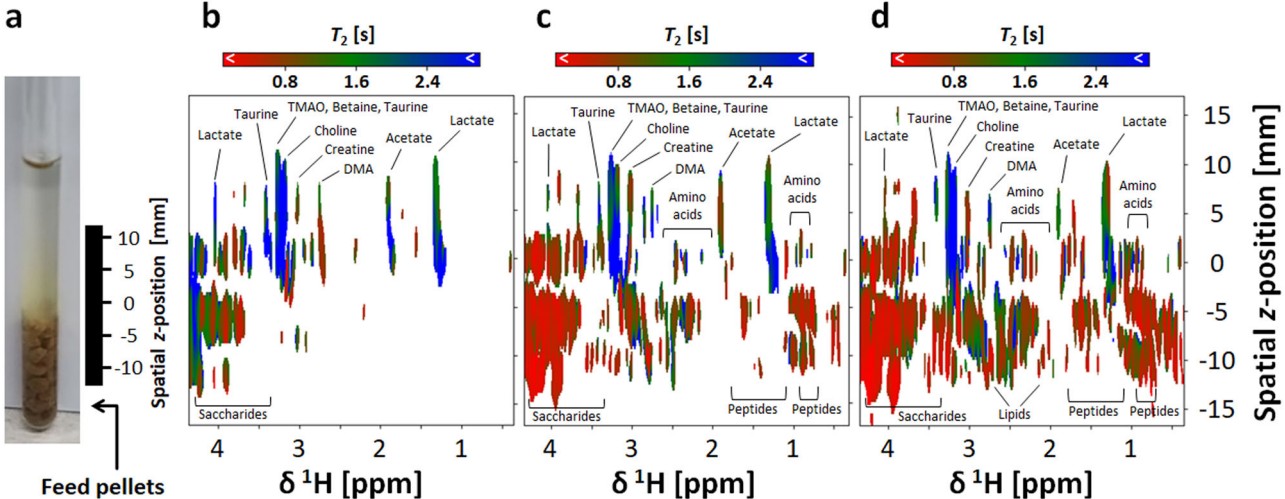

**Fig. 4 An example application of $T_2$-SMOOSY by solution-state NMR without MAS. a** Fish feed pellets and $D_2O$ were placed into 5-mm NMR tubes, and feed dissolution was evaluated. The black bars show the range of detected spatial $z$-position. **b–d** Pseudo-2D $T_2$-SMOOSY spectral images. The $T_2$-SMOOSY measurements were started approximately **b** 5 min, **c** 35 min, and **d** 65 min after sample preparation. The measurement times were approximately 5 min. The annotated major compounds are shown with relaxation times colored from red (short relaxation time) to blue (long relaxation time) according to the range shown in the bar above the figure. $T_2$-weighted parameters were repetition time (TR) = 600 ms, TE = 1 ms, number of excitations (NEX) = 2, and matrix 1024 × 16 (chemical shift × z-position).

improved the performance of compound diffusion evaluation in the spatial $z$-position. Moreover, if the diffusion encoding is changed to $T_1$ or $T_2$, the magnetic relaxation phenomena of the compounds in the spatial $z$-position can be analyzed, which is the first report at this time. The encoding in SMOOSY can also be changed with a new pulse sequence such as REST, and can be easily encoded to another pulse sequence for observing motility. For DOSY and ROSY analysis, the many processing methods of gradient strength or relaxation decay dimension for collecting the diffusion coefficient and relaxation time and/or separating signals have been used and developed in various studies. Broadly speaking, the two categories for this are inverse Laplace transform (ILT) and exponential fitting methods. However, ILT is considered to be an ill-defined approach, and exponential fitting is much safer for calculations of diffusion coefficients and relaxation time. An exponential fitting method also contains the problem of how to define the number of components when fitting for multiple exponents[41–43]. In this study, a single exponential curve fitting was selected because we did not aim for multi-component separation and wished to obtain a weighted image like MRI. It was thought that this selected result for a developed SMOOSY processor provides very useful information in processing of pseudo-3D SMOOSY spectra. The findings of this study showed that the proposed SMOOSY method was able to characterize the diffusion or relaxation of components of an intact biological sample spatially and in HR, as demonstrated in the analysis of the shrimp. Further evaluation of the differences and similarities in the diffusion phenomena of the compounds in the spatial $z$-position would be possible by statistical analysis such as PCA performed in this study. It would not have been possible to obtain these results for an intact sample using conventional methods. Applying this method to the living biological sample such as plants, seeds, animals, eggs, etc. could potentially facilitate the elucidation of unknown compound dynamics. As shown in results using solution-state NMR, SMOOSY can also be easily applied to other heterogeneous materials or mixed samples, and can also be used to evaluate these fluidity and flexibility for each region. However, when a heterogeneous sample is used in solution-state NMR, signal broadening and low sensitivity become problems. Recently, a method has been proposed to

suppress signal broadening due to inhomogeneous magnetic susceptibility in solution-state NMR[34–36]; if this method could be incorporated into SMOOSY, it may be possible to apply SMOOSY to a wide range of heterogeneous samples without using MAS. However, a limitation of the proposed method is the long measurement time. In this study using intact shrimp, the CSI experiment required around 1 h, and the $D$-SMOOSY experiment required around 16 h. In particular, $T_1$-SMOOSY needs a long relaxation wait time required around 36 h (Supplementary Table 3). The experiment with fish feed shows that reducing the number of measurement points and scans, $T_2$-SMOOSY can be measured in about 5 min. This time frame is sufficient for applications wherein data must be collected over a short period. However, such use is likely to involve a sacrifice of the spatial resolution and detection sensitivity. Recently, the ultrafast[44] approach to 2D NMR has been developed because of the short experimental time compared with traditional 2D NMR. Pseudo-2D DOSY and ROSY experiments can be also applied for this approach and are expected to be a potentially useful method for mixture analysis[45,46]. The low signal detection sensitivity is known as drawback of ultrafast approach; however, if the SMOOSY experiment can be applied to this approach as a high-speed measurement, the problem of a long-time experiment will be solved and will be a more powerful tool for mixture analysis of intact samples. Alternatively, when it is not necessary to evaluate the entire spatial $z$-position, and only partial evaluation is required, spatial selection is considered to be effective[46]. In addition, evaluation can be expected in a short time by using diffusion or relaxation as constant parameters and using them in weighted 2D experiments.

## Methods

**Sample preparation**. Natural samples of the shrimp *Palaemon sp.* were collected from an intertidal area at the mouth of Tsurumi River in Yokohama City, Kanagawa, Japan (35° 29′ 51.1″ N, 139° 40′ 34.5″ E). The shrimps, which were approximately 1.2 cm long, were immersed in methanol in a conical tube, which was placed in a desiccator for approximately 12 h under reduced pressure for immobilization. This treatment prevents the elution of components under the influence of the enzymes present in the shrimp. A $D_2O$ buffer with 4% paraformaldehyde was also tried for immobilization, but this did not completely prevent the elution of the components. After immobilization, the antennae and legs were

removed, and the sample was inserted with MeOD buffer into a ZrO$_2$ rotor with a 4-mm diameter for HR-MAS measurements. The inner lid was inserted into the rotor to fix the spatial $z$-position of the biological sample and to prevent MeOD solvent from leaking. For feed elution experiments, approximately 50 mg of Himezakura fish pellets (HIGASHIMARU Co., Ltd., Kagoshima, Japan) were placed into 5-mm NMR tubes after freeze drying. Thereafter, 500 μL of 0.1 M phosphate buffer solution [0.1 M K$_2$HPO$_4$/KH$_2$PO$_4$ (KPi); pH 7.0]/D$_2$O buffer with 1 mM sodium 2,2-dimethyl-2-silapentane-5-sulfonate-d6 (DSS-d6) was added. $T_2$-SMOOSY experiments began 5 min after sample preparation.

**NMR spectroscopy and pulse sequence details for experiments using intact shrimp**. 2D CSI[15,16] and pseudo-3D SMOOSY spectra of intact shrimp were recorded using an Avance III HD-500 instrument (Bruker Corp., Billerica, MA) equipped with a triple-resonance 4.0-mm HR-MAS probe with a $z$-axis gradient operating at 500.13 MHz for $^1$H and at 125.76 MHz for $^{13}$C. Supplementary Figure 1 and Supplementary Table 1 present the detailed pulse sequences shown in Fig. 1 for the CSI and SMOOSY. The 2D CSI and pseudo-3D SMOOSY spectra were recorded by HR-MAS at a MAS frequency of 3 kHz. When using this MAS frequency, the spinning sidebands appeared at 10.8 ppm and -1.2 ppm on the $^1$H spectrum; it was considered that there was no overlap with compound signals. In addition, to evaluate the results affected by faster or slower MAS frequencies, $D$-SMOOSY experiments at two different MAS frequencies were performed. One was the 2 kHz MAS frequency, in which the spinning sidebands appeared at 8.8 ppm and 0.8 ppm. The other was the 6 kHz MAS frequency, for which the spinning sidebands did not appear within spectral width. The pulse program parameters for 2D CSI and pseudo-3D SMOOSY are presented in Supplementary Table 3. The experimental temperature of these NMR experiments using HR-MAS was unified to 299 K without being affected by MAS. Water signal was suppressed by the pre-saturation pulse. The maximum gradient strength for this probe was 48.15 G/cm. The decoupling in the presence of scalar interactions 2 (DIPSI-2)[47] spin lock pulse for REST-SMOOSY was changed to Malcolm Levitt 17 sequence (MLEV-17)[48]. The 180° selective refocusing pulse (REBURP) was employed for the band selective pulse, and the olefin group in fatty acids with peaks in the range 6.0–4.5 ppm was excited. The gradient strength ratio for spatial $z$-profiling was defined according to the *diff2* list in the pulse programs. The *diff2* list controls the imaging gradient strength; the measured signals on indirect dimension constructed by increment gradient strength are transformed from the measurement space to the real space (spatial $z$-position) by Fourier transform. The experimental parameters for CSI and SMOOSY using intact shrimp are shown in the Methods section, Supplementary Tables 3 and 4.

**Solution-state NMR experiments for a heterogeneous system**. The solution-state NMR spectra of a heterogeneous system of fish feed pellets in KPi/D$_2$O were recorded using an Avance NEO-700 spectrometer (Bruker, Billerica, MA) equipped with an inverse triple-resonance cryogenic probe with a $z$-axis gradient for 5-mm diameter samples; this system was operated at 700.15 MHz for $^1$H and 176.06 MHz for $^{13}$C. $T_2$-SMOOSY spectra were recorded at 298 K. The analysis was performed to evaluate fish feed dissolution. For this, 16 complex F2 ($^1$H) points, 1024 complex F3 ($^1$H) points, and six points of gradient strength on F1 were recorded from two scans per F1 and F2 increment. The spectral width obtained for F3 was 16 ppm. The maximum gradient strength for F2 was 1.2 G/cm and the minimum was −1.2 G/cm. The increments were at equal intervals. Six increments in the variable counter list used for F1 were 25, 50, 75, 100, 125, and 150 loop counters. Direct (F3) and indirect (F2) dimensions were zero-filled to 2048 and 128 points, respectively. Details of experimental parameters are shown in Supplementary Tables 5 and 6.

**Data processing**. All the measured NMR data were first processed using Topspin 3 software. The 2D CSI spectra were produced by Fourier transformations of two dimensions (using *xfb*) and baseline correction (using *absd2* and *absd1*). The pseudo-3D SMOOSY spectra were produced by Fourier transformation of the chemical shift and spatial $z$-position axes (*ftnd*) and baseline correction (*abnd*) (Supplementary Fig. 2). The Topspin software processing parameters for 2D CSI and pseudo-3D SMOOSY are given in Supplementary Table 4. Projection of the pseudo-3D results onto the three end planes can be used to produce results equivalent to those from 2D DOSY, 2D diffusion-weighted imaging, and 2D CSI (Supplementary Fig. 2b, d, e, respectively). Sliced 1D $^1$H spectra for different body components on CSI spectrum from pseudo-3D SMOOSY spectrum show the spectral quality (Supplementary Fig. 2f). However, it can be difficult to obtain information visually and intuitively from 3D data. In MRI, diffusion and relaxation phenomena are generally presented as 2D enhanced images, which can be easily understood visually and intuitively. We therefore thought it would be most appropriate to convert the measurement data from pseudo-3D SMOOSY into pseudo-2D CSI diffusion- or relaxation-weighted spectral image. So, the processing of the diffusion or relaxation dimension and the generation of the pseudo-2D SMOOSY spectral image were performed using the processing tool named SMOOSY processor we developed, which was written in python 2. The ILT and exponential curve fitting are widely used for processing the diffusion or relaxation dimension in DOSY, IR, SR, CPMG, and PROJECT spectra. In this study,

SMOOSY processor used mono-exponential curve fitting without signal separation to produce simple spectra, similar to those in MRI. The details of processing for pseudo-3D SMOOSY spectra by SMOOSY processor are shown in the Supplementary Methods and Supplementary Fig. 3.

The 2D CSI spectra with annotated compounds are shown in Supplementary Fig. 6. The processed 2D CSI and pseudo-2D SMOOSY spectral images are shown in Fig. 2 and Supplementary Fig. 9. The processed data matrix exported by SMOOSY processor was used for feature extraction by PCA using the *prcomp* function in R software. PCA is used in this study to identify the significant compound diffusion or relaxation variances for each region. This analysis was considered that further evaluation of the difference and similarity in the diffusion phenomena of the compounds in the spatial $z$-position would be possible from pseudo-2D spectral image. Before performing PCA, it was necessary to normalize the signal strength of the 2D CSI spectrum[16] due to the gradient uniformity problem. If the gradient coil generates a perfectly uniform field, a perfect square will be obtained. However, gradient coils usually generate non-uniform fields with the gradient strength being stronger in the middle and weaker at the edges. Therefore, the scaling factor of the signal intensity at each spatial $z$-position was calculated from the CSI spectrum of the homogeneous sample (Supplementary Fig. 18). This scaling factor was normalized by multiplying the signal intensity at each spatial $z$-position of the CSI spectrum of the shrimp sample. Also, for correct the relative volumes of the parts of the intact shrimp, the signal intensities were normalized by total intensity was to be one at each spatial $z$-position.

## Data availability

All data supporting the findings of this study are available within the paper and its Supplementary Information files, and from the corresponding authors upon reasonable request.

## Code availability

The NMR pulse programs used in this study are available to obtain from Supplementary Data 1–6. The executable file or original python code of SMOOSY processor is available from http://dmar.riken.jp/Rscripts/.

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

## Acknowledgements
We thank Dr. M. Mekuchi (FRA) for advice of sample preparation. RIKEN Advanced Center for Computing and Communication provided use of a supercomputer, HOKU-SAI. This work was partially supported by the Agriculture, Forestry and Fisheries Research Council (to JK) as well as by the Strategic Innovation Program from Cabinet Office of Japan.

## Author contributions
K.I. and J.K. designated the study. K.I. and Y.T. collected the spectral data. K.I. and Y.T. performed the spectral analysis. K.I. developed the pulse programs and the analytical tool. All authors contributed to write the manuscript.

## Competing interests
The authors declare no competing interests.
