## [Peer Review File · Communications Chemistry]

Reviewers' comments:

Reviewer #1 (Remarks to the Author):

Ito et al. present a novel approach for identifying metabolites as a function of position within an animal using high resolution magic angle spinning NMR spectroscopy.

The method presented exploits the different characteristics of a given signal of the metabolites - the mobility (diffusion or relaxation properties) - and combined with TOCSY and advanced processing extracts the individual component spectra and thus identify the metabolites and their location in the sample.

The method is potentially very useful and this paper should be published in *commschem* if the authors can address a number of concerns and issues as highlighted below.

1. Mobility correlation spectra is not a term used generally in the NMR literature and this should be fully introduced at an early stage.
2. The authors should make it clear from the outset that the imaging is only 1D, not 3D. This is implied by the use of 'z-position' but not explicitly stated as would be required for a general audience.
3. 'NMR' is not an instrument but a method (actually 'NMR spectroscopy' is the method) - and this should be reflected on line 38
4. The authors undersell the impact of MRI and MRS (Nobel prize 1990) but also fail to define these terms.
5. DOSY and ROSY are pseudo-2D approaches, not 2D approaches as they do not require double Fourier transform.
6. Although relaxation relates to the correlation time for rotation of molecules, it is far-fetched to call this mobility and I am uncomfortable with the use of the term mobility to encompass diffusion and relaxation.
7. The description of how DOSY works contains a number of errors - for example gradients are not used to observe diffusion. Diffusion coefficients can be extracted from pulsed field gradient-attenuation profile collected in DOSY experiments.
8. The ultra-fast DOSY approaches from the group of Jean-Nicolas Dumez are relevant here and should be mentioned and referenced.
9. The authors use 'continuous wave method' where 'presaturation' is the more common term for this water suppression technique. They should explain why there are no methanol signals in their spectra (fully deuterated?).
10. Z-space is language from MR Imaging that many readers may be unfamiliar with.
11. The 'not alive shrimp' made me laugh - there may be a better way to describe it though.
12. The authors should comment on the MAS frequency and how faster/slower rotation would affect the results.
13. The references to supplementary figures are not introduced in a chronological order.
14. Some of the assignments of metabolites are optimistic but the methods are consistent with standard methods. The authors should comment on their confidence in the HSQC assignments.
15. H side and T side can be replaced with head and tail, or possibly head end and tail end.
16. Was any correction for the relative volumes of the parts of the shrimp used during the data analysis?
17. Although units of ' $\log(m^2/s)$ ' are found in the literature it is much clearer to report m^2/s and include a $\times 10^9$ or $\times 10^{10}$.
18. It is not clear why the DOSY signal for glycine was not obtained.
19. Insufficient reference is made to a standard set of experiments showing the results for a homogenised sample.
20. Are there any effects from the imaging gradients on the Stejskal-Tanner equation and diffusion measurements (or adverse effects on coherence transfer pathway selection for the ROSY based experiments)?

21. How do PCA and Parafac compare for analysis of results of this type?
22. The long measurement time is referred to - what was it? mention this is the main paper.
23. Was the shrimp body damaged by the measurement (spinning) process?
24. Bruker specific terms are used in the methods section - these need to be explained further - what does a 'diff2 list' do?
25. Figures should have cm / mm as units in z, not AU. This is not implemented consistently across all figures in the manuscript/SI.
26. Did the MAS rate affect the temperature and was the temperature calibrated?
27. Inverse Laplace Transform is an ill-defined approach and the mono-exponential fitting is much safer. The authors should discuss this and why they selected to use monoexponential fitting for all datasets in this study.
28. In figure 1 the 'composition' should be relabelled 'chemical shift'
29. The quality of the reproduction of the figures is poor in the review copy and should be checked.
30. The methods do not measure 'position-structure-mobility'. At least structure should be changed to chemical shift, but also it is safer to change mobility to the relevant property.
31. It is unhelpful to keep the same colour but change the meaning in the figures - I suggest using 2 colour schemes, one for position and the other for diffusion / relaxation.
32. It is unclear what 'green' means in fig 3 - the label suggests the middle of the sample but realistically it means the whole of the shrimp.
33. Are you sure that the effects observed are not due to volume effects and the amount of 'shell' at the different positions in the sample. What influence does the shell have on the relaxation properties?
34. In the CSI profiles, why do the different metabolites have different cutoffs at the top and bottom? The head and tail positions (or coil positions) of the shrimp do not seem well defined in the spectra. Is this effect real or is there an effect from the weighting in the Fourier transform in the CSI dimension that produces additional 'sinc wiggle' signals or broadening at the top/bottom for strong signals.?

The authors are to be commended for including the full pulse sequences for this research. The report is generally well written but would benefit from some additional proof-reading to correct some minor errors in language usage.

Reviewer #2 (Remarks to the Author):

The study introduces and evaluates the improved NMR-experiment(s) upon a recently explored HR-MAS CSI (spatially based metabolic profiling) by implementing a series of different 'spin-mobility-based pulse-experiments' prior to the detection with HR-MAS CSI (the authors denoted as HR-X-SMOOSY).

Indeed, these experiments could be vital in NMR-based metabolic profiling of intact organism. Aside from the ability of localized profiling with HR-MAS CSI, SMOOSY also permits to discriminate the molecular mobility (by T1, T2 and Diffusion) in an organism together with the spatial information for the longitudinal body axis.

The manuscript provides detail technical information (i.e. supplementary data including the pulse-sequences) offering a useful experimental basis for designing NMR profiling on an intact organism. However, the overall discussions – in both main text and supplementary sections – are targeted to NMR experts. It may be difficult to decipher for the non-experts. Generalizing the text could enhance the visibility in Communications Chemistry.

Some Comments and concerns:

-Despite the HR-X-SMOOSY can offer multiple molecular information (i.e. spatial and mobility) in a one-single NMR experiment, the integrity of the sample is questionable. One would expect a fast spinning (3000 Hz) would expect strong centrifugal stress upon the body, and perturb the metabolic profiles either by displacing the metabolite contents (especially the small metabolites) within the body, and/or enhancing the post-mortem activity. Compound to these effect from the fast sample spinning, the experimental time was carried out over 1-day long. The authors should address (or comment on) these issues.

-similar to the previous comment, one would expect that the fast-spinning (with long duration) would extract some biofluid contents (probably constitutes with small metabolites) from the body. (in fact, the use of the MeOD solvent in the rotor may increase the chances of the fluid extractions from the intact body). For example, the uniform diffusion coefficients for betaine and taurine (in Fig 2) across the body could be from the extracted fluid and not from the internal body structure.

-does the sample spinning affect the determined diffusion coefficients in HR-D-SMOOSY, even for macromolecules like the lipids?

-The discussions on the results of T1 and T2-weighted SMOOSY are lack of. For examples, in Figure 5 in the supplementary, the difference between figures d & e, and f & g. Based on the description, these spectra were deduced from 3D SMOOSY experiments with F1 dimension as relaxation (T1 or T2) measurements; if this is the case, then one could also extract the individual relaxation for each observable NMR metabolites?

-Also, the T1 and T2- weighted parameters of the spectra in Fig 5 are not clearly stated.

-All the 3D SMOOSY experiments (with the spin-evolution of diffusion, T1 and T2 in F1) have long acquisition time; I suppose this can be readily a 2D experiment with a constant parameter (for diffusion, T1 and T2) as a weighted 2D experiment.

Reviewer #3 (Remarks to the Author):

The authors present an interesting combination of HR-CSI and diffusion/relaxation-ordered spectroscopy applied to HR-MAS experiments of an intact shrimp. However, in its current form, the manuscript does not clearly demonstrate the novelty and relevance of the presented technique. As the authors point out themselves, combinations of CSI and DOSY have been implemented before [reference 18], so in terms of methodological approach, the method and the accompanying pulse sequences are not novel. The novelty is mainly in the application of this method biological samples. The presented application of this method to study intact bodies might be of scientific interest and I would expect the manuscript to focus on this.

However, the authors do not clearly show what new information can be gained with this method in the context of intact bodies or other biologically relevant samples. In particular for the relaxation based experiments, the results of which (p.5, l.172) are discussed only very briefly. A large amount of the text is concerned with specific experimental details and an in-depth description of the SMOOSY processor, which essentially provides the basic processing routines for these type of experiments which could be discussed in the supplementary material.

Finally the authors mention that long experiment times (i.e. limited sensitivity) is a limitation of the method but than opportunistically state two solutions, being ultrafast MAS and DNP NMR without giving them any thought considering their suitability of intact bodies. Ultrafast MAS is limited to minute volumes and the spinning speeds might well damage the body under study. DNP requires either freezing the entire sample or a way to add a polarized compound from a dissolution DNP setup, how could one implement that to study intact bodies?

In view of these comments I recommend major revisions, in particular to expand on the analysis of the results and to demonstrate the added value of combining HR-CSI with DOSY/ROSY in the

context of intact bodies and/or other relevant biological samples.

Minor comments:

* The manuscript is riddled with acronyms, which are not all well explained (e.g. DW-MRSI) which makes the paper difficult to read.

* The term MOSY seems to refer to an NMR experiment to measure electrophoretic mobilities (Kevin F. Morris and Charles S. Johnson, Jr., Mobility-Ordered Two-Dimensional Nuclear Magnetic Resonance Spectroscopy, JACS 1991) and does not apply to the kind of experiments described here.

* Also the need for a new acronym (SMOOSY) is not particularly clear in this case. On page 3 it says SMOOSY is the combination of the HR-CSI and MOSY sequences. The next section, however, talks about HR-SMOOSY, which implies SMOOSY is a combination of CSI and MOSY. But since MOSY is not a well-defined NMR experiment, the type of experiment combined with MOSY has to be added to the acronym as well (e.g., HR-REST1-SMOOSY). That means SMOOSY just becomes a synonym for CSI as HR-REST1-SMOOSY could just as well be called HR-REST1-CSI.

* On page 7 lists of gradient strengths and experimental settings are given which are not particularly relevant for the rest of the article and should be moved to the supplementary.

* Page 9 describes in some detail the LM method for non-linear least squares fitting, this is quite well known and can be left out.

* Equations 2-5 are the standard equations for diffusion and relaxation curves, and FOV. They can be found in most NMR books and do not need to be provided in the main text as no parameters related to these equations are clearly reported as results from this analysis.

* The Supplementary Information is rather extensive and contains subjects which are not mentioned in the main article, such as the analysis of chitin. The supplementary Information should be used to provide additional details to topics discussed in the main text.

Points-by-points response to Reviewer #1

Ito et al. present a novel approach for identifying metabolites as a function of position within an animal using high resolution magic angle spinning NMR spectroscopy.

The method presented exploits the different characteristics of a given signal of the metabolites - the mobility (diffusion or relaxation properties) - and combined with TOCSY and advanced processing extracts the individual component spectra and thus identify the metabolites and their location in the sample.

The method is potentially very useful and this paper should be published in Commschem if the authors can address a number of concerns and issues as highlighted below.

1. Mobility correlation spectra is not a term used generally in the NMR literature and this should be fully introduced at an early stage.

[Response. 1-1]

Receiving your mention, we also thought the sentence “mobility correlation spectra” is not used as general expression.

So, this sentence was modified in revised manuscript, as follows;

Page 2 line 3 in abstract section,

“the nuclear magnetic resonance (NMR) data including spatial z-position, chemical shift, and diffusion or relaxation.”

Page 20 line 6 in Fig .1 caption

“This method provides the information including spatial z-position from 1D-imaging, chemical shift, and diffusion or relaxation.”

2. The authors should make it clear from the outset that the imaging is only 1D, not 3D. this is implied by the use of 'z-position' by not explicitly stated as would be required for a general audience.

[Response. 1-2]

The imaging dimension used in this study was made clear, as follows:

Page 4 line 22, in the introduction section

“The CSI and SMOOSY used in this study had only information of the spatial z-position from one-dimensional (1D) imaging and did not include the x- and y-positions from three-dimensional (3D) imaging like the MRI.”

3. 'NMR' is not an instrument but a method (actually 'NMR spectroscopy' is the method) - and this should be reflected on line 38

[Response. 1-3]

We modified the word “instruments” to “methods” in page 3 line 1 (total line was 38 in previous manuscript).

4. The authors undersell the impact of MRI and MRS (Nobel prize 1990) but also fail to define these terms.

[Response. 1-4]

We now define the terms of MRI and MRS (page3 line 2).

In addition, two great contributions were cited, as follows:

Page 3 line 14-21

“In this background, the Nobel Prize in Chemistry 1991 was awarded to Richard R. Ernst “for his contributions to the development of the methodology of high-resolution nuclear magnetic resonance (NMR) spectroscopy”¹². This technology has become a very powerful tool in the identification of molecular structures in organic chemistry. It also has been put to practical use in MRI in the medical field and has been a powerful clue for diagnostic imaging. Furthermore, the Nobel Prize in Physiology or Medicine 2003 was awarded to Paul Lauterbur and Peter Mansfield “for their discoveries concerning magnetic resonance imaging”¹².”

We also added the following reference:

12. Kauffman, G. Nobel prize for MRI imaging denied to Raymond V. Damadian a decade ago. *Chem. Educator* **19**, 73–90 (2014).

5. *DOSY and ROSY are pseudo-2D approaches, not 2D approaches as they do not require double Fourier transform.*

[Response. 1-5]

We modified the sentences concerned with your suggestion as follows:

Page 3 line 31

“Diffusion-ordered spectroscopy (DOSY) is a pseudo two-dimensional (2D) NMR technique”

Page 3 line 35- page 4 line 3

“Relaxation-ordered spectroscopy (ROSY) such as saturation recovery (SR), inversion recovery (IR), Carr–Purcell–Meiboom–Gill (CPMG), periodic refocusing of J evolution by coherence transfer (PROJECT), and relaxation-encoded selective total correlation spectroscopy (REST) are pseudo 2D NMR techniques”

SMOOSY is also a “pseudo 3D” NMR technique; the appropriate sentence was modified.

The text on these expressions had been thoroughly checked in manuscript and supporting information.

6. *Although relaxation relates to the correlation time for rotation of molecules, it is far-fetched to call this mobility and I am uncomfortable with the use of the term mobility to encompass diffusion and relaxation.*

[Response 1-6]

In compliance with this comment, we changed the word “mobility” and “mobilities” to “diffusion or relaxation” throughout in manuscript and supporting information. Also, the name of the new pulse sequence was changed from “spatial mobility ordered spectroscopy” to “spatial molecular-dynamically ordered spectroscopy”.

7. *The description of how DOSY works contains a number of errors - for example gradients are not used to observe diffusion. Diffusion coefficients can be extracted from pulsed field gradient-attenuation profile collected in DOSY experiments.*

[Response. 1-7]

The sentence describing how DOSY works was modified as follows:

Page 3 line 31-33

“Diffusion-ordered spectroscopy (DOSY) is a pseudo two-dimensional (2D) NMR technique that diffusion coefficients can be extracted from pulsed field gradient-attenuation profile collected in DOSY experiments.”

We also fixed minor errors related to this throughout the manuscript.

8. *The ultra-fast DOSY approaches from the group of Jean-Nicolas Dumez are relevant here and should be mentioned and referenced.*

[Response. 1-8]

We added the following references about ultra-fast DOSY and ROSY approaches:

41. Shrot, Y. & Frydman, L. Single-scan 2D DOSY NMR spectroscopy. *J. Magn. Reson.* **195**, 226–231 (2008). [10.1016/j.jmr.2008.09.011](https://doi.org/10.1016/j.jmr.2008.09.011), Pubmed:18835796

42. Dumez, J. N. Spatial encoding and spatial selection methods in high-resolution NMR spectroscopy. *Prog. Nucl. Magn. Reson. Spectrosc.* **109**, 101–134 (2018). [10.1016/j.pnmrs.2018.08.001](https://doi.org/10.1016/j.pnmrs.2018.08.001), Pubmed:30527133

Discussion about this was also added, as follows:

Page 11 line 32- page 11 line 7

“Recently, the ultrafast⁴⁰ approach to 2D NMR has been developed because of the short experimental time compared with traditional 2D NMR. Pseudo 2D DOSY and ROSY experiments can be also applied for this approach and are expected to be a potentially useful method for mixture analysis^{41,42}. The low signal detection sensitivity is known as drawback of ultrafast approach; however, if the HR-SMOOSY experiment can be applied to this approach as a high-speed measurement, the problem of a long-time experiment will be solved and will be a more powerful tool for mixture analysis of intact samples. Alternatively, when it is not necessary to evaluate the entire spatial z -position and only partial evaluation is required, spatial selection is considered to be effective⁴². In addition, evaluation can be expected in a short time by using diffusion or relaxation as constant parameters and using them in weighted 2D experiments.”

9. The authors use 'continuous wave method' where 'presaturation' is the more common term for this water suppression technique. The should explain why there are no methanol signals in their spectra (fully deuteriated?).

[Response. 1-9]

The phrase “continuous wave,” “CW” was replaced with “presaturation”, as follows:

Page 5 line 7, and page 13 line 4 in manuscript

“water signal was suppressed by the presaturation pulse”

Page 9 line 8 in Supplementary Information

“The presat denotes water presaturation”

Accordingly, “CW” was replaced with “presat” in Fig 1 and Supplementary Figure 1.

Methanol as a fixative and solvent appeared at 3.37 ppm ($\delta^1\text{H}$) in HR-HSQC, HR-CSI, and HR-SMOOSY spectra, not fully deuteriated. Methanol signal is shown in **Supplementary Figure 6**, and the chemical shift is listed in **Supplementary Table 2**. The annotation No. is 71; however, **Fig. 2** showed only annotated major metabolites of the shrimp in our previous draft. Therefore, methanol was added and annotated in **Fig. 2**. We also added the following to the text:

Page 7 line 32 - 34

“However, methanol as a fixative and solvent appeared at 3.37 ppm. The signals of metabolites appeared nearby here may have affected the diffusion coefficient by signal overlap.”

10. Z-space is language from MR Imaging that many readers may be unfamiliar with.

[Response. 1-10]

We agree that “z-space” is not used commonly. However, the language “z-position” is used for a broad audience in many journals. We therefore replaced all mentions of “z-space” with “spatial z-position.”

11. The 'not alive shrimp' made me laugh - there may be a better way to describe it though.

[Response. 1-11]

We modified the sentence “not alive shrimp” to “shrimp with tissue fixed in methanol” in Page 6 line 9.

12. The authors should comment on the MAS frequency and how faster/slower rotation would affect the results.

[Response. 1-12]

The additional experiments were performed, and the results and discussion were added as follows:

Page 5 line 26 - page 6 line 2

“The effect of different MAS on results was evaluated in HR-DOSY experiments using homogeneous sample (**Supplementary Figure 5**). In the HR-DOSY experiment, a comparison of the pseudo 2D DOSY spectrum measured at 3 kHz with the MAS frequency and the pseudo 2D DOSY spectrum measured at 6 kHz with the MAS frequency showing the change in the diffusion coefficient. The diffusion coefficient when the MAS frequency was 6 kHz was approximately 3.08×10^{-10} (m^2/s) larger than the diffusion coefficient when the MAS frequency was 3 kHz. From this result, a rapid MAS frequency affected the diffusion coefficient of compounds to be fast. In SS-NMR, it has been reported that the rate of MAS changes the T_1 and T_2 relaxation time^{25,26}. The relationship between MAS and diffusion coefficient or relaxation time was considered to be proportional. Therefore, although the diffusion coefficient and relaxation time changed depending on MAS, it was considered that the evaluation of the relative profile was also possible in DOSY, ROSY, and SMOOSY experiments.”

Also, the references were added.

25. Bakhmutov, V. I. Strategies for solid-state NMR studies of materials: from diamagnetic to paramagnetic porous solids. *Chem. Rev.* **111**, 530–562 (2011). [10.1021/cr100144r](https://doi.org/10.1021/cr100144r), Pubmed:20843066

26. Mroue, K. H. *et al.* Proton-Detected Solid-State NMR Spectroscopy of Bone with Ultrafast Magic Angle Spinning. *Sci. Rep.* **5**, 11991 (2015). [10.1038/srep11991](https://doi.org/10.1038/srep11991), Pubmed: 26153138

Page 9 line 10-17

“There is also concern that MAS may disrupt the internal metabolic profile. To evaluate this possibility, a non-rotating HR-MAS experiment was attempted, but analysis was difficult because most of the signals were broad. As a solution to this,

there was a recently reported pulse sequence of a HR NMR experiment using a heterogeneous sample in a non-rotating state³⁴⁻³⁶. Applying these pulse sequences to SMOOSY was expected to solve the concerns of the effects of MAS, and was considered to be a future issue. In the last section, an application example of the SMOOSY experiment of different samples by NMR in a non-rotating state without using HR-MAS is shown.”

Also, please see **Response 1-23** for comments on these issues.

13. *The references to supplementary figures are not introduced in a chronological order.*

[Response. 1-13]

The references to supplementary figures including additional data were reviewed and modified for to introduce them in chronological order.

14. *Some of the assignments of metabolites are optimistic but the methods are consistent with standard methods. The authors should comment on their confidence in the HSQC assignments.*

[Response. 1-14]

The NMR signals of metabolites in shrimp were annotated by referring from SpinAssign tool and a previous report of metabolites in shrimp. As additional analysis for signal assignment, chain assignment using correlation signals from ¹H-¹³C HSQC-TOCSY was employed to verify the validity of annotated signals on HSQC spectrum. The usefulness of the chain assignment of a metabolic mixture was reported by our previous study (Ito et al. 2016).

This result was added at page 6 lines 18-24 in the manuscript:

“Furthermore, ¹H-¹³C HSQC-total correlation spectroscopy (HSQC-TOCSY) was used to verify the validity of the annotated signals by chain assignment using TOCSY correlation signals. The usefulness of the chain assignment for metabolic mixture was reported by our previous study²⁸. TOCSY correlation signals of almost of all the annotated metabolites were confirmed (**Supplementary Figure 7**), and the assignment of signals of metabolites was considered to be reliable.”

This experimental detail was added at page 4 in the Supplementary Information.

“The HSQC-TOCSY analysis was performed to evaluate the validity of signal annotation. For this, 256 complex F1 (¹³C) and 2048 complex F2 (¹H) points were recorded from 64 scans per F1 increment. The spectral widths obtained for F1 and F2 were 140 ppm and 16 ppm, respectively. The mixing time using decoupling in the presence of scalar interactions 2 (DIPSI-2) as a spin lock pulse was 60 ms.”

Supplementary Figure 7 was added at page 15 in the Supplementary Information.

“**Supplementary Figure 7** ¹H-¹³C HR-HSQC spectrum of an intact shrimp (blue), ¹H-¹³C HSQC spectrum (red), and ¹H-¹³C HSQC-TOCSY spectrum of extract from powdered shrimp (gray) with a MeOD/HEPES buffer. HR-HSQC was measured using HR-MAS probe at a MAS frequency of 3 kHz at 299 K. The peak numbers represent the annotated chemical shifts by the SpinAssign tool in InterSpin⁷. (listed in **Supplementary Table 2**). The gray lines show the chemical fragment network of same spin system by chain assignment using correlation signals of HSQC-TOCSY. The fragment network of signals from cholesterol is not shown because of complex network.”

15. *H side and T side can be replaced with head and tail, or possibly head end and tail end.*

[Response. 1-15]

“H side” was replaced with “head,” and “T side” was replaced with “tail” throughout the manuscript. The concerned figures were modified.

16. *Was any correction for the relative volumes of the parts of the shrimp used during the data analysis?*

[Response. 1-16]

As we mention in a previous manuscript, the shrimp data was corrected at each spatial z-position before PCA and added to the PARAFAC as data analysis, as follows:

Page 14 line 13-23

“Before performing PCA, it was necessary to normalize the signal strength of the 2D HR-CSI spectrum¹⁶ due to the gradient uniformity problem. If the gradient coil generates a perfectly uniform field, a perfect square will be obtained. However, gradient coils usually generate non-uniform fields with the gradient strength being stronger in the middle and weaker at the

edges. Therefore, the scaling factor of the signal intensity at each spatial z -position was calculated from the HR-CSI spectrum of the homogeneous sample (**Supplementary Figure 18**). This scaling factor was normalized by multiplying the signal intensity at each spatial z -position of the HR-CSI spectrum of the shrimp sample. Also, for correct the relative volumes of the parts of the intact shrimp, the signal intensities were normalized by total intensity was to be 1 at each spatial z -position.”

For this explanation, the NMR experiment and results using homogeneous samples were added to the Supporting Information, as follows:

Page 2 in Supplementary Information

“Homogenous sample preparation for NMR experiments using HR-MAS

For nuclear magnetic resonance (NMR) experiments by high-resolution magic angle spinning (HR-MAS) using homogeneous gel sample, sucrose and alanine were prepared to a final concentration of 12.5 mM, agarose was prepared to 2% (w/v), and sodium 2,2-dimethyl-2-silapentane-5-sulfonate-d₆ (DSS-d₆) was prepared to a final concentration of 1 mM in deuterium oxide (D₂O) buffer. This sample was melted at 99°C, and the melted sample was put into the 4 mm HR-MAS rotor. After this, the sample in rotor was cooled at room temperature.”

Page 2-3 in Supplementary Information

“NMR experiments with HR-MAS using homogeneous sample

The spectra of homogeneous sample were recorded using an Avance III HD-500 instrument (Bruker Corp., Billerica, MA) equipped with a triple-resonance 4 mm HR-MAS probe with a z -axis gradient operating at 500.13 MHz for ¹H and at 125.76 MHz for ¹³C. HR-CSI, traditional pseudo two-dimensional (2D) HR diffusion-ordered spectroscopy (HR-DOSY), and pseudo three-dimensional (3D) HR-D-SMOOSY spectra of homogeneous sample were recorded at a MAS frequency of 3 kHz at 299 K. The HR-CSI analysis was performed to calculate scaling factor for normalize the signal intensity at each spatial z -position. The traditional HR-DOSY and D-SMOOSY analysis was performed to evaluate whether imaging gradients on the diffusion measurement affect to calculate the diffusion coefficients. In the HR-CSI experiment, 128 complex F1 (¹H) points, 2048 complex F2 (¹H) points were recorded from 24 scans per F1 increment. The spectral widths obtained for F2 was 17 ppm. The maximum gradient strength for F1 was 9.63 G/cm, and the minimum gradient strength for F1 was -9.63 G/cm, the 128 increment was equal intervals. The details of experimental parameters are shown in **Supplementary Tables 3 and 4**. In the traditional HR-DOSY experiment, 16,384 complex F2 (¹H) points and 16 points of gradient strength on F1 were recorded from 16 scans per F1 increment. The spectral widths obtained for F2 was 16 ppm. The 16 increments in the diffusion list used for F1 were 0.963, 3.948, 6.934, 9.919, 12.904, 15.89, 18.875, 21.86, 24.845, 27.831, 30.816, 33.801, 36.787, 39.772, 42.757, and 45.742 G/cm. The diffusion time was 60 ms. In the HR-D-SMOOSY experiment, 128 complex F1 (¹H) points, 2048 complex F3 (¹H) points, and 24 points of gradient strength on F2 were recorded from 16 scans per F1 and F2 increment. The spectral widths obtained for F3 was 16 ppm. The details of experimental parameters are shown in **Supplementary Tables 3**. 2D HR-CSI and pseudo 3D HR-D-SMOOSY spectra were processed by SMOOSY processor. The exported csv file including a matrix of signal intensities of 2D HR-CSI spectrum was used for calculate the scaling factors at each spatial z -position. Also, the exported csv file including a matrix of diffusion coefficients of pseudo 2D HR-D-SMOOSY spectral image was used for whether the diffusion coefficients were changed at each spatial z -position. To compare between traditional pseudo 2D HR-DOSY spectrum and sliced pseudo 2D HR-DOSY spectra of pseudo 3D HR-D-SMOOSY spectrum at two spatial z -points, the dimensions of gradient strengths were processed by exponential curve fitting using equation (2) on Dynamics Center 2.5 software.

The homogeneous sample was also applied to NMR experiments to investigate the effects of the different MAS frequencies to diffusion coefficient. The traditional pseudo 2D HR-DOSY spectra of homogeneous sample were recorded at a MAS frequency of 3 kHz and 6 kHz at 299 K. The traditional HR-DOSY experiment was performed by same NMR experimental parameters in previous paragraph.”

Page 14 line 21-23

“Also, for correct the relative volumes of the parts of the intact shrimp, the signal intensities were normalized by total intensity was to be 1 at each spatial z -position.”

17. Although units of 'log(m²/s)' are found in the literature it is much clearer to report m²/s and include a $\times 10^9$ or $\times 10^{10}$.

[Response. 1-17]

The units of 'log(m²/s)' were changed to $\times 10^9$ or $\times 10^{10}$ in page 5 of manuscript. However, in D-SMOOSY spectral image, logarithmic scale was considered more suitable for make the difference clear of diffusion coefficient at each spatial z -position than linear scale. The difference of spectral image between linear scale and logarithmic scale was added in **Supplementary Figure 3**.

18. It is not clear why the DOSY signal for glycine was not obtained.

[Response. 1-18]

This consideration was added at page 7 in manuscript, as follows:

“The signal intensity of glycine in intact shrimp was reduced rapidly upon gradient strength dimension. Optimization of the exponential curve fitting for the signal of glycine did not converge and failed, and this signal image on the HR-D-SMOOSY spectral image is not shown. The reason for this was considered to be that glycine was so quick to diffuse or may have a special motility because this metabolite is used for osmotic pressure adjustment. If experimental NMR parameters are set for target to metabolite with fast diffusion, this problem may be solved.”

19. Insufficient reference is made to a standard set of experiments showing the results for a homogenised sample.

[Response. 1-19]

The results for a homogenized sample were also modified according to **Response 1-16**.

20. Are there any effects from the imaging gradients on the Stejskal-Tanner equation and diffusion measurements (or adverse effects on coherence transfer pathway selection for the ROSY based experiments)?

[Response. 1-20]

We confirmed whether the imaging gradients affected the diffusion measurement and diffusion coefficient calculation by comparing these with traditional HR-DOSY and HR-D-SMOOSY experiments using homogeneous samples. When calculation of diffusion coefficient, the Stejskal-Tanner equation was not used but expression (2) in experimental section was used because encoded diffusion experiment for HR-D-SMOOSY was PBLED. This additional materials and methods are answered in **Response. 1-16**. The results about this concern were added in **Supplementary Figure 4**. The mention about this result was added in manuscript, as follows:

Page 4 line 19-25

“For the pulse sequence of HR-D-SMOOSY, we were concerned whether the imaging gradients in CSI affected the diffusion measurements and diffusion coefficient calculation. However, the traditional pseudo 2D HR-DOSY spectrum and sliced pseudo 2D HR-DOSY spectrum from the pseudo 3D HR-D-SMOOSY spectrum using a homogeneous sample produced the same results including diffusion coefficients; imaging gradients in CSI did not affect diffusion measurements or calculation of the diffusion coefficient (**Supplementary Figure 4**).”

21. How do PCA and Parafac compare for analysis of results of this type?

[Response. 1-21]

The parallel factor analysis (PARAFAC) as a feature extraction method using the pseudo 3D HR-SMOOSY spectrum was also employed for comparison with the result of PCA. This additional analysis was added as follows:

Page 6 line 1-5 in Supplementary Information

“PARAFAC of pseudo 3D HR-SMOOSY spectra of intact shrimp

The three-way parallel factor analysis (PARAFAC) facilitates extraction of the features from 3D data, for comparison the results with principal component analysis (PCA) using 2D data, PARAFAC using pseudo 3D HR-SMOOSY spectra was performed by parafac function in multiway package on R software.”

Page 9 line 30- page 10 line 2

“In addition, three-way parallel factor analysis (PARAFAC) of pseudo 3D HR-SMOOSY spectra was also performed and compared with the results of PCA using pseudo 2D HR-SMOOSY spectral images. The features of intact shrimp at each spatial z -position could be extracted, and the contributions of component 1 and 2 on PARAFAC scores were signal intensities of mainly metabolites (**Supplementary Figure 14-16**). Thus, PCA using the pseudo 2D HR-SMOOSY spectral image was more suitable for feature extraction of diffusion or relaxation of metabolites at each spatial z -position than PARAFAC using pseudo 3D HR-SMOOSY spectra.”

22. The long measurement time is referred to - what was it? mention this is the main paper.

[Response. 1-22]

The NMR experimental (measurement) time is shown in **Supplementary Table 3**.

The mention of NMR experimental time was added on page 11, line 29-32, as follows:

“In this study using intact shrimp, the HR-CSI experiment required around 1 h, and the HR-D-SMOOSY experiment required around 16 h. In particular, HR-T₁-SMOOSY needs a long relaxation wait time required around 36 h (**Supplementary Table 3**)”

23. *Was the shrimp body damaged by the measurement (spinning) process?*

[Response. 1-23]

We evaluated the damage of the shrimp body by MAS process. This mention was added in main text, as follows:

Page 8 line 31 - page 9 line 10

“The damage of the shrimp body by MAS process was evaluated. NMR experiments with MAS from 2 kHz to 6 kHz showed no body damage and no outflow of contents (**Supplementary Figure 11**). Similar to a recent report for living organism, MAS at 2.5 kHz for 1 hour is the highest speed that could be used without affecting survival¹⁴. Similar to another report, in the HR-CSI experiment using intact wasp, the wasp body structure was found to be intact after enduring a 4-hour span of rotation at 4 kHz¹⁶. However, it should be noted that faster MAS and using other sample types may have caused damage. In addition, as a result of comparing the pseudo 2D HR-D-SMOOSY spectral image measured by these different MAS, although the effect of individual differences was somewhat apparent, the pattern of signal appearance and imaging profile in spatial *z*-position was almost the same in experiments using different MAS. It was confirmed that the diffusion coefficient was slightly changed by different MAS, as mentioned in the experiment with the homogeneous sample. For example, the difference of diffusion coefficient of lipid when the MAS frequencies were 2 kHz and 6 kHz was approximately 9.54×10^{-10} (m²/s).”

24. *Bruker specific terms are used in the methods section - these need to be explained further - what does a 'diff2 list' do?*

[Response. 1-24]

The sentence for Aiming of *diff2* list was added in page 13 line 11–14, as follows:

“The *diff2* list controls the imaging gradient strength; the measured signals on indirect dimension constructed by increment gradient strength are transformed from the measurement space to the real space (spatial *z*-position) by Fourier transform.”

25. *Figures should have cm / mm as units in z, not AU. This is not implemented consistently across all figures in the manuscript/SI.*

[Response. 1-25]

Supplementary Figure 2, Supplementary Figure 6, and Additional **Supplementary Figure 4** do not have cm/mm as units in the *z*-position. These spectra were processed and drawn using TopSpin software, and the exact *z*-position was difficult to show using only TopSpin software without a SMOOSY processor. However, because these data had to be shown as a figure before being applied to the SMOOSY processor and displayed together with the scale of other data, the unit was set to A.U. Therefore, the sentence “This spectrum was spectrum before process by SMOOSY processor” was written in the description of the figures. The method of correction is described in **Supplementary Information**, and the results are reflected in **Supplementary Figure 9**.

26. *Did the MAS rate affect the temperature and was the temperature calibrated?*

[Response. 1-26]

The sentence about an experimental temperature using HR-MAS was added as follows:

Page 13 line 3-4

“The experimental temperature of these NMR experiments using HR-MAS was unified to 299 K without being affected by MAS.”

27. *Inverse Laplace Transform is an ill-defined approach and the mono-exponential fitting is much safer. The authors should discuss this and why they selected to use monoexponential fitting for all datasets in this study.*

[Response. 1-27]

The sentences about selection of mono-exponential fitting, as follows:

Page 11 line 3 -14

“For DOSY and ROSY analysis, the many processing methods of gradient strength or relaxation decay dimension for collecting the diffusion coefficient and relaxation time and/or separating signals have been used and developed in various studies. Broadly speaking, the two categories for this are inverse Laplace transform (ILT) and exponential fitting methods. However, ILT is considered to be an ill-defined approach, and exponential fitting is much safer for calculations of diffusion coefficients and relaxation time. An exponential fitting method also contains the problem of how to define the number of components when fitting for multiple exponents³⁷⁻³⁹. In this study, a single exponential curve fitting was selected because we did not aim for multi-component separation and wished to obtain a weighted image like MRI. It was thought that this selected result for a developed SMOOSY processor provides very useful information in processing of pseudo 3D HR-SMOOSY spectra.”

Also, the following new references were cited in the text:

37. Castañar, L. *et al.* The GNAT: A new tool for processing NMR data. *Magn. Reson. Chem.* **56**, 546–558 (2018). [10.1002/mrc.4717](https://doi.org/10.1002/mrc.4717), Pubmed:29396867

38. Yuan, B. *et al.* Reconstructing diffusion ordered NMR spectroscopy by simultaneous inversion of Laplace transform. *J. Magn. Reson.* **278**, 1–7 (2017). [10.1016/j.jmr.2017.03.004](https://doi.org/10.1016/j.jmr.2017.03.004), Pubmed:28301804

39. Cherni, A., Chouzenoux, E. & Delsuc, M. A. PALMA, an improved algorithm for DOSY signal processing. *Analyst* **142**, 772–779 (2017). [10.1039/c6an01902a](https://doi.org/10.1039/c6an01902a), Pubmed:28120953

28. *In figure 1 the 'composition' should be relabelled 'chemical shift'*

[Response. 1-28]

The “composition” in **Fig.1** was relabeled “chemical shift.”

29. *The quality of the reproduction of the figures is poor in the review copy and should be checked.*

[Response. 1-29]

The quality of the reproduction of all figures was checked and modified.

30. *The methods do not measure 'position-structure-mobilty'. At least structure should be changed to chemical shift, but also it is safer to change mobility to the relevant property.*

[Response. 1-30]

The “structure” was modified “chemical shift” for observation, and “metabolite” for evaluation throughout the manuscript. The “mobilty” was also modified according to **Response 1-1** and **1-6**.

31. *It is unhelpful to keep the same colour but change the meaning in the figures - I suggest using 2 colour schemes, one for position and the other for diffusion / relaxation.*

[Response. 1-31]

We also considered the colors in the figures are confusing. Therefore, the color scheme in **Fig. 3, Supplementary Figure 12**, and additional **Supplementary Figure 14** was modified.

32. *It is unclear what 'green' means in fig 3 - the label suggests the middle of the sample but realistically it means the whole of the shrimp.*

[Response. 1-32]

We clarified some differences by adding color to the color scheme.

33. *Are you sure that the effects observed are not due to volume effects and the amount of 'shell' at the different positions in the sample. What influence does the shell have on the relaxation properties?*

[Response. 1-33]

A sentence about this consideration was added as follows:

Page 7 line 17-27

“Furthermore, the difference in the diffusion of these metabolites at each spatial z-position of the intact shrimp is thought to be due to the volume effect and the amount and interaction of chitin constituting the shell. For example, in a paper studying

the molecular structure of fungal cell walls by SS-NMR, components closer to the cell membrane, such as chitin and glucan, had lower motility and were affected by relaxation time³³. In this study, SS-NMR was used to evaluate the chemical shift and relaxation time of chitin, which constitutes the shrimp shell (**Supplementary Figure 8**); however, it was impossible to evaluate chitin due to the problem of fast relaxation time in CSI using HR-MAS. If these macromolecules and low molecules can be evaluated by imaging at the same time, this can lead to new discoveries.”

Also, the following reference was added:

33. Kang, X. *et al.* Molecular architecture of fungal cell walls revealed by solid-state NMR. *Nat. Commun.* **9**, 2747 (2018). [10.1038/s41467-018-05199-0](https://doi.org/10.1038/s41467-018-05199-0), Pubmed:[30013106](https://pubmed.ncbi.nlm.nih.gov/30013106/)

34. In the CSI profiles, why do the different metabolites have different cutoffs at the top and bottom? The head and tail positions (or coil positions) of the shrimp do not seem well defined in the spectra. Is this effect real or is there an effect from the weighting in the Fourier transform in the CSI dimension that produces additional 'sinc wiggle' signals or broadening at the top/bottom for strong signals.?

[Response. 1-34]

The sentence addressing this consideration was added as follows:

Page 8 line 23-30

“The difference between the top and bottom shrimp in CSI and D-SMOOSY profiles as well as no noticeable difference in other HR-SMOOSY profiles was considered to be due to the dynamic range problem. Similar to a recent report, in the HR-CSI experiment using intact wasp by MAS at a frequency of 4 kHz, the profile was different between tail, mid, and head. By increasing the echo time (TE), signals with fast relaxation times disappeared, and buried signals appeared¹⁶. In this study, TE was unified to 1 ms to obtain an overall profile. However, a more different profile was expected to be observed by changing the length of TE.”

The spatial z-position of the shrimp head and tail were clearly defined in the spectrum.

Sync Wiggle by Fourier transform was not generated, but depending on the window function, this is generated.

Due to the baseline correction, there was no spread at the top/bottom of the strong signal. Before correction, there was some spread.

The authors are to be commended for including the full pulse sequences for this research. The report is generally well written but would benefit from some additional proof-reading to correct some minor errors in language usage.

Points-by-points response to Reviewer #2

The study introduces and evaluates the improved NMR-experiment(s) upon a recently explored HR-MAS CSI (spatially based metabolic profiling) by implementing a series of different 'spin-mobility-based pulse-experiments' prior to the detection with HR-MAS CSI (the authors denoted as HR-X-SMOOSY).

Indeed, these experiments could be vital in NMR-based metabolic profiling of intact organism. Aside from the ability of localized profiling with HR-MAS CSI, SMOOSY also permits to discriminate the molecular mobility (by T1, T2 and Diffusion) in an organism together with the spatial information for the longitudinal body axis.

The manuscript provides detail technical information (i.e. supplementary data including the pulse-sequences) offering a useful experimental basis for designing NMR profiling on an intact organism. However, the overall discussions – in both main text and supplementary sections – are targeted to NMR experts. It may be difficult to decipher for the non-experts. Generalizing the text could enhance the visibility in Communications Chemistry.

[Response. 2-1]

The text of the manuscript has been overhauled and revised to be more general. However, we believe it is difficult to make all text into more general expressions while serving all reviewers. Compared with specialized NMR papers, the pulse program itself is open to the public and simple processing tools are provided, so non-NMR specialists can easily use it and are expected to be applicable in a wide range of fields. In addition, compared with HR-MAS, a commonly used SMOOSY experiment in solution NMR, was also added. These additional experiments are described as follows:

Page 3 line 23-33 in Supplementary Information

“For membrane filtration experiments using D-SMOOSY, alanine, sucrose, and 140 mg polyvinyl alcohol (PVA; YAMATO Co., LTD., Tokyo, Japan) were suspended in 0.1 M KPi/D₂O buffer with 1 mM DSS-d₆; alanine and sucrose were prepared to a final concentration of 12.5 mM in 1 mL buffer. Four sheets of polytetrafluoroethylene (PTFE) membrane filter (Merck KGaA, Darmstadt, Germany) with a 1 μm pore size were stacked and put into the 5 mm NMR tube so as to be about 2 cm from the bottom. After this, prepared sample was put on the membrane filter, and NMR measurements were started after 9 h. For feed diffusion experiment using D-SMOOSY, approximately 50 mg Himezakura (HIGASHIMARU Co., LTD., Kagoshima, Japan) as fish feed pellets were put into the 5 mm NMR tube after freeze drying. And 500 μL 0.1 M KPi/D₂O buffer with 1 mM DSS-d₆ was poured in the 5 mm NMR tube. NMR measurements were started after 9 h.”

Page 4 line 29 – page 5 line 2 in Supplementary Information

“D-SMOOSY spectra were recorded at 298 K for fish feed sample and PTFE membrane sample as other application examples. The D-SMOOSY analysis was performed to evaluate the fish feed diffusion and PTFE membrane filtration. For this, 32 complex F1 (¹H) points, 2048 complex F3 (¹H) points, and 14 points of gradient strength on F2 were recorded from 16 scans per F1 and F2 increment. The spectral widths obtained for F3 was 12 ppm. The maximum gradient strength for F1 was 2.41 G/cm, and the minimum gradient strength for F1 was -2.41 G/cm, the 32 increment was equal intervals. The 14 increments in the diffusion list used for F2 were 0.963, 4.408, 7.852, 11.297, 14.741, 18.186, 21.63, 25.075, 28.52, 31.964, 35.409, 38.853, 42.298, and 45.743 G/cm. The diffusion time was 60 ms.”

Sentences about the experimental results were also added as follows:

Page 10 line 4-28

“**SMOOSY experiments by solution-state NMR.** To investigate the further applicability of the SMOOSY experiment, two different samples were used to evaluate the diffusion of components at the z-position by D-SMOOSY. These experiments were performed with solution-state NMR, which is more commonly used than HR-MAS. The first was a membrane filtration experiment (**Supplementary Figure 17a**). In this experiment, a deuterium oxide (D₂O) solution containing polyvinyl alcohol (PVA), alanine, and sucrose was prepared, a polytetrafluoroethylene (PTFE) membrane was fixed in an NMR tube, and components in the solution permeated the membrane. From the pseudo 2D SMOOSY spectrum image, it can be seen that PVA having a large molecular weight and a low diffusion coefficient is accumulated at the top of the membrane and hardly passes through it. Alanine and sucrose have permeated the membrane, and that their diffusion coefficients are different. The low molecular weight PVA becomes acetate. This has a fast diffusion coefficient and passes through the membrane. It was possible to evaluate whether or not it permeated through the membrane based on the molecular weight and the diffusion coefficient. For this experiment, it was thought that it may be applied to the performance evaluation of a membrane used for sewage treatment and filtration. The second was a fish feed diffusion experiment (**Supplementary Figure 17b**). This is an experiment in which feed pellets were put into the bottom of an NMR tube, D₂O was poured in, and the elution phenomenon of components was evaluated. Peptides have a low diffusion coefficient and remain at the bottom of the NMR tube, whereas small molecules such as amino acids, organic acids, and saccharides have large diffusion coefficients and are eluted into the solution and diffused widely. For this experiment, this was expected to lead to research on feed development. Details of these experiments can be found in the **Supplementary Information**.”

We believe these additional experiments will be understandable and interesting to non-NMR specialists.

Some Comments and concerns:

-Despite the HR-X-SMOOSY can offer multiple molecular information (i.e. spatial and mobility) in a one-single NMR experiment, the integrity of the sample is questionable. One would expect a fast spinning (3000 Hz) would expect strong centrifugal stress upon the body, and perturb the metabolic profiles either by displacing the metabolite contents (especially the small metabolites) within the body, and/or enhancing the post-mortem activity. Compound to these effect from the fast sample spinning, the experimental time was carried out over 1-day long. The authors should address (or comment on) these issues.

[Response. 2-2]

The additional experiments were performed, and the results and discussion were also added as follows:

Page 5 line 26-page 6 line 2

“The effect of different MAS on results was evaluated in HR-DOSY experiments using homogeneous sample (**Supplementary Figure 5**). In the HR-DOSY experiment, a comparison of the pseudo 2D DOSY spectrum measured at 3 kHz with the MAS frequency and the pseudo 2D DOSY spectrum measured at 6 kHz with the MAS frequency showing the change in the diffusion coefficient. The diffusion coefficient when the MAS frequency was 6 kHz was approximately 3.08×10^{-10} (m²/s) larger than the diffusion coefficient when the MAS frequency was 3 kHz. From this result, a rapid MAS frequency affected the diffusion coefficient of compounds to be fast. In SS-NMR, it has been reported that the rate of MAS changes the T₁ and T₂ relaxation time^{25,26}. The relationship between MAS and diffusion coefficient or relaxation time was considered to be proportional. Therefore, although the diffusion coefficient and relaxation time changed depending on MAS, it was considered that the evaluation of the relative profile was also possible in DOSY, ROSY, and SMOOSY experiments.”

Also, the references were added.

25. Bakhmutov, V. I. Strategies for solid-state NMR studies of materials: from diamagnetic to paramagnetic porous solids. *Chem. Rev.* **111**, 530–562 (2011). [10.1021/cr100144r](https://doi.org/10.1021/cr100144r), Pubmed:[20843066](https://pubmed.ncbi.nlm.nih.gov/20843066/)

26. Mroue, K. H. *et al.* Proton-Detected Solid-State NMR Spectroscopy of Bone with Ultrafast Magic Angle Spinning. *Sci. Rep.* **5**, 11991 (2015). [10.1038/srep11991](https://doi.org/10.1038/srep11991), Pubmed: [26153138](https://pubmed.ncbi.nlm.nih.gov/26153138/)

Page 9 line 10-17

“There is also concern that MAS may disrupt the internal metabolic profile. To evaluate this possibility, a non-rotating HR-MAS experiment was attempted, but analysis was difficult because most of the signals were broad. As a solution to this, there was a recently reported pulse sequence of a HR NMR experiment using a heterogeneous sample in a non-rotating state³⁴⁻³⁶. Applying these pulse sequences to SMOOSY was expected to solve the concerns of the effects of MAS, and was considered to be a future issue. In the last section, an application example of the SMOOSY experiment of different samples by NMR in a non-rotating state without using HR-MAS is shown.”

Also, please see the next **Response 2-3** for comments on these issues.

-similar to the previous comment, one would expect that the fast-spinning (with long duration) would extract some biofluid contents (probably constitutes with small metabolites) from the body. (in fact, the use of the MeOD solvent in the rotor may increase the chances of the fluid extractions from the intact body). For example, the uniform diffusion coefficients for betaine and taurine (in Fig 2) across the body could be from the extracted fluid and not from the internal body structure.

[Response. 2-3]

We evaluated the damage of the shrimp body by MAS process. This was mentioned in the main text as follows:

Page 8 line 31-page 9 line 10

“The damage of the shrimp body by MAS process was evaluated. NMR experiments with MAS from 2 kHz to 6 kHz showed no body damage and no outflow of contents (**Supplementary Figure 11**). Similar to a recent report for living organism, MAS at 2.5 kHz for 1 hour is the highest speed that could be used without affecting survival¹⁴. Similar to another report, in the HR-CSI experiment using intact wasp, the wasp body structure was found to be intact after enduring a 4-hour span of rotation at 4 kHz¹⁶. However, it should be noted that faster MAS and using other sample types may have caused damage. In addition, as a result of comparing the pseudo 2D HR-D-SMOOSY spectral image measured by these different MAS, although the effect of individual differences was somewhat apparent, the pattern of signal appearance and imaging profile in spatial z-position was almost the same in experiments using different MAS. It was confirmed that the diffusion coefficient was slightly changed by different MAS, as mentioned in the experiment with the homogeneous sample. For example, the difference of diffusion coefficient of lipid when the MAS frequencies were 2 kHz and 6 kHz was approximately 9.54×10^{-10} (m²/s).”

-does the sample spinning affect the determined diffusion coefficients in HR-D-SMOOSY, even for macromolecules like the lipids?

[Response. 2-4]

Comments on this issue are included in **Responses 2-2** and **2-3**, so please check the answers above.

-The discussions on the results of T1 and T2-weighted SMOOSY are lack of. For examples, in Figure 5 in the supplementary, the difference between figures d & e, and f & g. Based on the description, these spectra were deduced from 3D SMOOSY experiments with F1 dimension as relaxation (T1 or T2) measurements; if this is the case, then one could also extract the individual relaxation for each observable NMR metabolites?

[Response. 2-5]

We added to the Results and Discussion section about HR-T₂-SMOOSY, HR-T₁-SMOOSY, and HR-REST-SMOOSY as follows:

Page 8 line 8-30

“With HR-T₂-SMOOSY, the lipid profile was similar to that of HR-D-SMOOSY, with a shorter relaxation time at the head and a longer relaxation time at the tail. TAG signals with small diffusion coefficients and short relaxation times could not be captured with sufficient quality. Thus, the exponential curve fitting was not well performed in relaxation decay due to fast signal decay. In contrast, glycine, which had a large diffusion coefficient and could not be captured by HR-D-SMOOSY, could be evaluated with both HR-T₂-SMOOSY and HR-T₁-SMOOSY. REST allows the extraction of component sub-spectra from mixtures (**Supplementary Figure 10**). Therefore, HR-REST-SMOOSY was considered to also allow the extraction of component sub-spectra from mixtures at each spatial z-position.

In this study, the HR-REST-SMOOSY experiment focused on shrimp DHA and EPA. As a result, the relaxation time was shorter at the head and longer at the tail. REST-SMOOSY was considered useful for evaluating individual metabolites while avoiding signal overlap. HR-T₁-SMOOSY and HR-REST₁-SMOOSY were not as pronounced as other HR-SMOOSY techniques used in this study; however, this experiment may be useful for other samples. The difference between the top and bottom shrimp in CSI and D-SMOOSY profiles as well as no noticeable difference in other HR-SMOOSY profiles was considered to be due to the dynamic range problem. Similar to a recent report, in the HR-CSI experiment using intact wasp by MAS at a frequency of 4 kHz, the profile was different between tail, mid, and head. By increasing the echo time (TE), signals with fast relaxation times disappeared, and buried signals appeared¹⁶. In this study, TE was unified to 1 ms to obtain an overall profile. However, a more different profile was expected to be observed by changing the length of TE.”

-Also, the T1 and T2- weighted parameters of the spectra in Fig 5 are not clearly stated.

[Response. 2-6]

The T1 and T2- weighted parameters were added in the description of **Supplementary Figure 9**.

-All the 3D SMOOSY experiments (with the spin-evolution of diffusion, T1 and T2 in F1) have long acquisition time; I suppose this can be readily a 2D experiment with a constant parameter (for diffusion, T1 and T2) as a weighted 2D experiment.

[Response. 2-7]

We also thought 3D SMOOSY experiments may be used as weighted 2D experiments. This was mentioned at page 11 line 29-page 12 line 7,

“In this study using intact shrimp, the HR-CSI experiment required around 1 h, and the HR-D-SMOOSY experiment required around 16 h. In particular, HR-T₁-SMOOSY needs a long relaxation wait time required around 36 h (**Supplementary Table 3**). Recently, the ultrafast⁴⁰ approach to 2D NMR has been developed because of the short experimental time compared with traditional 2D NMR. Pseudo 2D DOSY and ROSY experiments can be also applied for this approach and are expected to be a potentially useful method for mixture analysis^{41,42}. The low signal detection sensitivity is known as drawback of ultrafast approach; however, if the HR-SMOOSY experiment can be applied to this approach as a high-speed measurement, the problem of a long-time experiment will be solved and will be a more powerful tool for mixture analysis of intact samples. Alternatively, when it is not necessary to evaluate the entire spatial z-position and only partial evaluation is required, spatial selection is considered to be effective⁴². In addition, evaluation can be expected in a short time by using diffusion or relaxation as constant parameters and using them in weighted 2D experiments.”

Points-by-points response to Reviewer #3

The authors present an interesting combination of HR-CSI and diffusion/relaxation-ordered spectroscopy applied to HR-MAS experiments of an intact shrimp. However, in its current form, the manuscript does not clearly demonstrate the novelty and relevance of the presented technique.

As the authors point out themselves, combinations of CSI and DOSY have been implemented before [reference 18], so in terms of methodological approach, the method and the accompanying pulse sequences are not novel. The novelty is mainly in the application of this method biological samples. The presented application of this method to study intact bodies might be of scientific interest and I would expect the manuscript to focus on this.

However, the authors do not clearly show what new information can be gained with this method in the context of intact bodies or other biologically relevant samples. In particular for the relaxation based experiments, the results of which (p.5, l.172) are discussed only very briefly. A large amount of the text is concerned with specific experimental details and an in-depth description of the SMOOSY processor, which essentially provides the basic processing routines for these type of experiments which could be discussed in the supplementary material.

[Response. 3-1]

The novelty of this research is the development of pulse programs combining ROSY and REST with CSI. In many cases, phantoms have been used to develop imaging techniques, but in this study, the use of actual biological samples is also novel and applicable. Furthermore, the development of applications that enhance the readability of the experimental results is also new. Compared with specialized and difficult NMR papers, this also clarifies how to publish and use pulse programs and simple analysis applications, making it more conscious of non-NMR specialists. Another point is that this is not just a mere introduction of techniques. For this reason, the technical aspects of the methods and results are particularly emphasized. However, because the results and considerations for the newly developed mitigation experiment were hardly described in the main text, these related sentences were also added to the main text as follows:

Page 8 line 6–30

“Therefore, the other pseudo 2D HR-SMOOSY spectral image was also evaluated. This showed differences in the relaxation times of each metabolite in the spatial z -position (**Supplementary Figure 9**). With HR- T_2 -SMOOSY, the lipid profile was similar to that of HR-D-SMOOSY, with a shorter relaxation time at the head and a longer relaxation time at the tail. TAG signals with small diffusion coefficients and short relaxation times could not be captured with sufficient quality. Thus, the exponential curve fitting was not well performed in relaxation decay due to fast signal decay. In contrast, glycine, which had a large diffusion coefficient and could not be captured by HR-D-SMOOSY, could be evaluated with both HR- T_2 -SMOOSY and HR- T_1 -SMOOSY. REST allows the extraction of component sub-spectra from mixtures (**Supplementary Figure 10**). Therefore, HR-REST-SMOOSY was considered to also allow the extraction of component sub-spectra from mixtures at each spatial z -position.

In this study, the HR-REST-SMOOSY experiment focused on shrimp DHA and EPA. As a result, the relaxation time was shorter at the head and longer at the tail. REST-SMOOSY was considered useful for evaluating individual metabolites while avoiding signal overlap. HR- T_1 -SMOOSY and HR-REST $_1$ -SMOOSY were not as pronounced as other HR-SMOOSY techniques used in this study; however, this experiment may be useful for other samples. The difference between the top and bottom shrimp in CSI and D-SMOOSY profiles as well as no noticeable difference in other HR-SMOOSY profiles was considered to be due to the dynamic range problem. Similar to a recent report, in the HR-CSI experiment using intact wasp by MAS at a frequency of 4 kHz, the profile was different between tail, mid, and head. By increasing the echo time (TE), signals with fast relaxation times disappeared, and buried signals appeared¹⁶. In this study, TE was unified to 1 ms to obtain an overall profile. However, a more different profile was expected to be observed by changing the length of TE.”

Also, as mentioned, the application of the method is of interest to the reader. So, we added an application of a different two-sample method. This additional experiment will expand the possibilities of this method, and we hope that it will attract the interest of researchers in a wider range of fields.

This is described in [Response. 3-3].

Finally the authors mention that long experiment times (i.e. limited sensitivity) is a limitation of the method but than opportunistically state two solutions, being ultrafast MAS and DNP NMR without giving them any thought considering their suitability of intact bodies. Ultrafast MAS is limited to minute volumes and the spinning speeds might well damage the body under study. DNP requires either freezing the entire sample or a way to add a polarized compound from a dissolution DNP setup, how could one implement that to study intact bodies?

[Response. 3-2]

Due to an error in the description of DNP, text related to this was deleted. In addition, “Ultrafast” described in this paper refers to the pulse sequence that enables high-speed measurement, not the rotation speed with high rate of the MAS. For these reasons, the last discussion had been amended as follows:

Page 11 line 29-page 12 line 7

“In this study using intact shrimp, the HR-CSI experiment required around 1 h, and the HR-D-SMOOSY experiment required around 16 h. In particular, HR-T₁-SMOOSY needs a long relaxation wait time required around 36 h (**Supplementary Table 3**). Recently, the ultrafast⁴⁰ approach to 2D NMR has been developed because of the short experimental time compared with traditional 2D NMR. Pseudo 2D DOSY and ROSY experiments can be also applied for this approach and are expected to be a potentially useful method for mixture analysis^{41,42}. The low signal detection sensitivity is known as drawback of ultrafast approach; however, if the HR-SMOOSY experiment can be applied to this approach as a high-speed measurement, the problem of a long-time experiment will be solved and will be a more powerful tool for mixture analysis of intact samples. Alternatively, when it is not necessary to evaluate the entire spatial z-position and only partial evaluation is required, spatial selection is considered to be effective⁴². In addition, evaluation can be expected in a short time by using diffusion or relaxation as constant parameters and using them in weighted 2D experiments.”

The following articles were also referenced for the discussion of “Ultrafast.”

40. Giraudeau, P., Shrot, Y. & Frydman, L. Multiple ultrafast, broadband 2D NMR spectra of hyperpolarized natural products. *J. Am. Chem. Soc.* **131**, 13902–13903 (2009). [10.1021/ja905096f](https://doi.org/10.1021/ja905096f), Pubmed:[19743849](https://pubmed.ncbi.nlm.nih.gov/19743849/)

41. Shrot, Y. & Frydman, L. Single-scan 2D DOSY NMR spectroscopy. *J. Magn. Reson.* **195**, 226–231 (2008). [10.1016/j.jmr.2008.09.011](https://doi.org/10.1016/j.jmr.2008.09.011), Pubmed:[18835796](https://pubmed.ncbi.nlm.nih.gov/18835796/)

42. Dumez, J. N. Spatial encoding and spatial selection methods in high-resolution NMR spectroscopy. *Prog. Nucl. Magn. Reson. Spectrosc.* **109**, 101–134 (2018). [10.1016/j.pnmrs.2018.08.001](https://doi.org/10.1016/j.pnmrs.2018.08.001), Pubmed:[30527133](https://pubmed.ncbi.nlm.nih.gov/30527133/)

In view of these comments I recommend major revisions, in particular to expand on the analysis of the results and to demonstrate the added value of combining HR-CSI with DOSY/ROSY in the context of intact bodies and/or other relevant biological samples.

[Response. 3-3]

Following the reviewers' comments, many additional experiments were performed, and the manuscript was significantly revised. Among these additional experiments, the SMOOSY experiment was applied to other samples. In particular, there are many experiments using biological samples in imaging, and experiments on HR-CSI using biological samples have recently been reported. Therefore, it was considered necessary to show examples of the application of different systems and to make the content interesting to readers in a wider field. Therefore, this paper presents an applied example of the SMOOSY experiment using two newly added samples. This experiment is a solution-state NMR experiment more commonly used than HR-MAS. The content is also conscious of non-NMR specialists, and we hope that it will lead to various advanced research. The results of additional experiments were added as follows:

Page 10 line 4-28

“**SMOOSY experiments by solution-state NMR.** To investigate the further applicability of the SMOOSY experiment, two different samples were used to evaluate the diffusion of components at the z-position by D-SMOOSY. These experiments were performed with solution-state NMR, which is more commonly used than HR-MAS. The first was a membrane filtration experiment (**Supplementary Figure 17a**). In this experiment, a deuterium oxide (D₂O) solution containing polyvinyl alcohol (PVA), alanine, and sucrose was prepared, a polytetrafluoroethylene (PTFE) membrane was fixed in an NMR tube, and components in the solution permeated the membrane. From the pseudo 2D SMOOSY spectrum image, it can be seen that PVA having a large molecular weight and a low diffusion coefficient is accumulated at the top of the membrane and hardly passes through it. Alanine and sucrose have permeated the membrane, and that their diffusion coefficients are different. The low molecular weight PVA becomes acetate. This has a fast diffusion coefficient and passes through the membrane. It was possible to evaluate whether or not it permeated through the membrane based on the molecular weight and the diffusion coefficient. For this experiment, it was thought that it may be applied to the performance evaluation of a membrane used for sewage treatment and filtration. The second was a fish feed diffusion experiment (**Supplementary Figure 17b**). This is an experiment in which feed pellets were put into the bottom of an NMR tube, D₂O was poured in, and the elution phenomenon of components was evaluated. Peptides have a low diffusion coefficient and remain at the bottom of the NMR tube, whereas small molecules such as amino acids, organic acids, and saccharides have large diffusion coefficients and are eluted into the solution and diffused widely. For this experiment, this was expected to lead to research on feed development. Details of these experiments can be found in the **Supplementary Information**.”

Minor comments:

* *The manuscript is riddled with acronyms, which are not all well explained (e.g. DW-MRSI) which makes the paper difficult to read.*

[Response. 3-4]

Revised definitions for all abbreviations and corrected as explained.

** The term MOSY seems to refer to an NMR experiment to measure electrophoretic mobilities (Kevin F. Morris and Charles S. Johnson, Jr., Mobility-Ordered Two-Dimensional Nuclear Magnetic Resonance Spectroscopy, JACS 1991) and does not apply to the kind of experiments described here.*

[Response. 3-5]

Following the suggestions of the reviewers, we also considered the term MOSY was not suitable for NMR experiments in this study. The term MOSY had simply been replaced by the terms DOSY and ROSY.

** Also the need for a new acronym (SMOOSY) is not particularly clear in this case. On page 3 it says SMOOSY is the combination of the HR-CSI and MOSY sequences. The next section, however, talks about HR-SMOOSY, which implies SMOOSY is a combination of CSI and MOSY. But since MOSY is not a well-defined NMR experiment, the type of experiment combined with MOSY has to be added to the acronym as well (e.g., HR-REST1-SMOOSY). That means SMOOSY just becomes a synonym for CSI as HR-REST1-SMOOSY could just as well be called HR-REST1-CSI.*

[Response. 3-6]

The sentences concerning this were modified. For example as follows:

Page 2 line 4-6

“This experiment was collectively named high-resolution spatial molecular-dynamically ordered spectroscopy (HR-SMOOSY).”

Page 4 line 31-36

“The method referred to here as HR-SMOOSY, was driven by diffusion encoding (HR-D-SMOOSY, as well as HR-CSI-DOSY), T1 encoding (HR-T1-SMOOSY, as well as HR-CSI-SR), T2 encoding (HR-T2-SMOOSY, as well as HR-CSI-PROJECT), REST₁ encoding (HR-REST1-SMOOSY, as well as HR-CSI-REST₁), or REST₂ encoding (HR-REST2-SMOOSY, as well as HR-CSI-REST₂).”

** On page 7 lists of gradient strengths and experimental settings are given which are not particularly relevant for the rest of the article and should be moved to the supplementary.*

[Response. 3-7]

The lists of gradient strengths and experimental settings in manuscript were moved to Supplementary Information.

** Page 9 describes in some detail the LM method for non-linear least squares fitting, this is quite well known and can be left out.*

[Response. 3-8]

Describes in some detail the LM method in manuscript were moved to Supplementary Information.

** Equations 2-5 are the standard equations for diffusion and relaxation curves, and FOV. They can be found in most NMR books and do not need to be provided in the main text as no parameters related to these equations are clearly reported as results from this analysis.*

[Response. 3-9]

The standard equations for diffusion and relaxation curves, FOV, and its description in main text were moved to Supplementary Information.

** The Supplementary Information is rather extensive and contains subjects which are not mentioned in the main article, such as the analysis of chitin. The supplementary Information should be used to provide additional details to topics discussed in the main text.*

[Response. 3-10]

The supplementary information, supplementary figures and tables all supported the main text, but some were not well linked. Therefore, the supplemental information and main text had been reviewed to provide additional details for the topics described in the main text.

Reviewers' comments:

Reviewer #1 (Remarks to the Author):

The authors have been thorough in addressing the comments of all reviewers and I am happy to recommend publication of this paper.

Reviewer #2 (Remarks to the Author):

The revised manuscript has appropriately addressed all the reviewer's comments and questions. Overall, the manuscript presents a detail NMR methodological description of advancing the HR-MAS CSI experiments.

Due to the added spin-edited (spin-relaxations and molecular diffusions) experiments prior to the CSI acquisition, it has significantly lengthened the experimental time (<16hr); and compound this with the fast sample rotation (even at 3kHz), the experiments probably have distorted the natural-state of the metabolism in the organism. In general, long acquisition time and fast rotation have long been an issue in metabolic profiling with HR-MAS NMR, and many efforts – from slow sample spinning to stationary condition (with iSQC mentioned in the text), and with ultrafast 2D acquisition – have been attempt to address these.

Unlike in the starndard HR-MAS profiling, aside from the integrity of the metabolism, displacement of the metabolites during the sample rotation is also of concern in CSI experiment. To overcome this issue, this study fixed (immobilized) the body tissue with methanol – this reflects the white rigid body tissue shown in Fig S11. However, chemical fixation is not commonly applied for profiling the metabolome in biospecimens. This is because the results may lead to biased interpretations of the metabolic activities. In fact, how does this affect the diffusion and relaxation results as compared to the normal state 'fresh' tissue (even in term of the relative profile mentioned in the text). The authors should address the overall effects of the chemical fixation on the metabolome.

The study should acknowledge a very recent report (in Analyst, DOI: 10.1039/d0an00118j) on HR-MAS CSI, which has 'partially' addressed both of these issues – sample integrity and metabolic displacement – with slow sample rotation (500 Hz) experiment, and with only one-hour acquisition on a non-chemically fixed intact organism.

Since spectral quality is a key component to metabolic profiling, the authors should consider showing the spectral quality of a 1D chemical shift spectrum (either with a standard 1D CPMG, or better yet, the individual slice F2 projection of HR-CSI of the different body component).

Reviewer #3 (Remarks to the Author):

The authors have made significant improvements to their manuscript. This greatly improved readability and I recommend publication. However, I do think the authors should consider including some of the more important figures from the supplementary material in the main text and possibly condense the descriptions of the experiments in the main text. Condensing the main text but including SI figures, while giving more detailed descriptions in the SI, will make the manuscript more accessible to a broader audience.

Points-by-points response to Reviewer #2

The revised manuscript has appropriately addressed all the reviewer's comments and questions. Overall, the manuscript presents a detail NMR methodological description of advancing the HR-MAS CSI experiments.

Due to the added spin-edited (spin-relaxations and molecular diffusions) experiments prior to the CSI acquisition, it has significantly lengthened the experimental time (<16hr); and compound this with the fast sample rotation (even at 3kHz), the experiments probably have distorted the natural-state of the metabolism in the organism. In general, long acquisition time and fast rotation have long been an issue in metabolic profiling with HR-MAS NMR, and many efforts – from slow sample spinning to stationary condition (with iSQC mentioned in the text), and with ultrafast 2D acquisition – have been attempt to address these.

Unlike in the starndard HR-MAS profiling, aside from the integrity of the metabolism, displacement of the metabolites during the sample rotation is also of concern in CSI experiment. To overcome this issue, this study fixed (immobilized) the body tissue with methanol – this reflects the white rigid body tissue shown in Fig S11. However, chemical fixation is not commonly applied for profiling the metabolome in biospecimens. This is because the results may lead to biased interpretations of the metabolic activities. In fact, how does this affect the diffusion and relaxation results as compared to the normal state 'fresh' tissue (even in term of the relative profile mentioned in the text). The authors should address the overall effects of the chemical fixation on the metabolome.

[Response. 2-1]

NMR experiments using long-term, high-speed rotation likely affect metabolic profiling of biological samples. The intact shrimp used in this study was not a live and fresh sample. Further, the scope of application in our study is not limited to biological samples, and it does not focus only on metabolites. Rather the physical properties in various sample systems were our primary targets. Thus, as the use of terms “metabolite” and “metabolic dynamics” are not inappropriate, and we now used the expressions such as “component” or “compound.”

We have also revised the title as follows:

“Spatial molecular-dynamically ordered NMR spectroscopy of intact bodies and heterogeneous systems”

HR-MAS was not used other than the biological sample.

Therefore, “HR” was omitted from the experimental name “HR-SMOOSY” to eliminate the narrower connotation.

We also examined whether a rapid SMOOSY experiment (5 min) was possible.

These additions and corrections are as follows:

Page 4 lines 26–30

“As examples of possible applications, we described comprehensive analyses of an intact biological sample and two heterogeneous systems such as the diffusion of feeding stimulants from aquaculture feeds and transport of small molecules via a membrane filter. We also present an example of spectrum analysis using the SMOOSY processor.”

Page 10 line 25 to Page 11 line 17

“Stimulants in aquaculture feeds, such as betaine, organic acids, amines, amino acids, saccharides, and their mixtures, are important for the development of new aquaculture feeds³⁸⁻⁴⁰. Therefore, the second example was fish feed diffusion (**Supplementary Figure 17b**). Feed pellets were placed at the bottom of an NMR tube, D₂O was added, and the diffusion of components of the feed was evaluated. Peptides have a low diffusion coefficient, e.g. ca. 6.81×10^{-11} (m²/s) and remain at the bottom of the NMR tube. Small molecules, such as amino acids, organic acids, and saccharides, have large diffusion coefficients; they dissolve and diffuse widely. Especially, the diffusion coefficients for lactate and acetate were large, ca. 1.74×10^{-9} (m²/s). Thus, the diffusion of several compounds into the feed was significant with rapid dissolution. To detect the dissolution of compounds into the feed, high-speed measurements immediately after sample preparation is necessary.

Thereafter, T₂-SMOOSY was adjusted to about 5 min per measurement. The measurement was started at an early stage after sample preparation, and the dissolution process of the compounds into the feed was assessed (**Fig. 4**). The dissolution of lactate and acetate was confirmed with relaxation times of about 1.61 s (**Fig. 4b**). In addition, the dissolution of TMAO, betaine, taurine, choline, creatine, dimethylamine (DMA), and saccharides was confirmed, whereas the dissolution of amino acids was confirmed 35 min after sample preparation (**Fig. 4c**). A peptide signal was also detected. This finding might be explained by the prolonged relaxation time that was due to the affinity with water or dissolution to improve mobility. After 65 min of sample preparation (**Fig. 4d**), dissolution and affinity with water proceeded further. These compounds in solution

are likely to attract fish. T_2 -SMOOSY over a short time frame may have problems with deterioration of spatial resolution and detection sensitivity; however, it is sufficient for the preliminary evaluation of samples. These results indicate that the technique is applicable to research on feed development including the exploration of attractant compounds for fishes. Details of the above experiments are present in the **Supplementary Information**.”

38. Carlberg, H., Cheng, K., Lundh, T. & Brannas, E. Using self-selection to evaluate the acceptance of a new diet formulation by farmed fish. *Appl. Anim. Behav. Sci.* **171**, 226–232 (2015). [10.1016/j.applanim.2015.08.016](https://doi.org/10.1016/j.applanim.2015.08.016)

39. Hill, H. A., Trushenski, J. T. & Kohler, C. C. Utilization of soluble canola protein concentrate as an attractant enhances production performance of sunshine bass fed reduced fish meal, plant-based diets. *J. World Aquacult. Soc.* **44**, 124–132 (2013). [10.1111/jwas.12005](https://doi.org/10.1111/jwas.12005)

40. Kasumyan, A. O. & Doving, K. B. Taste preferences in fishes. *Fish Fish.* **4**, 289–347 (2003). [10.1046/j.1467-2979.2003.00121.x](https://doi.org/10.1046/j.1467-2979.2003.00121.x)

Page 12 lines 23–27

The experiment with fish feed shows that reducing the number of measurement points and scans, T_2 -SMOOSY can be measured in about 5 min. This time frame is sufficient for applications wherein data must be collected over a short period. However, such use is likely to involve a sacrifice of the spatial resolution and detection sensitivity.”

Page 13 lines 16–21

“For feed dissolution experiments, approximately 50 mg of Himezakura fish feed pellets (HIGASHIMARU Co., Ltd., Kagoshima, Japan) were placed into 5-mm NMR tubes after freeze-drying. Thereafter, 500 μ L of 0.1 M phosphate buffer solution [0.1 M K_2HPO_4/KH_2PO_4 (KPi); pH 7.0]/ D_2O buffer with 1 mM sodium 2,2-dimethyl-2-silapentane-5-sulfonate-d6 (DSS-d6) was added. T_2 -SMOOSY measurements began 5 min after sample preparation.”

Page 14 lines 16–29

“**Solution-state NMR experiments for a heterogeneous system.** The solution-state NMR spectra of a heterogeneous system of fish feed pellets in KPi/ D_2O were recorded using an Avance NEO-700 spectrometer (Bruker, Billerica, MA) equipped with an inverse triple-resonance cryogenic probe with a z -axis gradient for 5-mm diameter samples; this system was operated at 700.15 MHz for 1H and 176.06 MHz for ^{13}C . T_2 -SMOOSY spectra were recorded at 298 K. The analysis was performed to evaluate fish feed dissolution. For this, 16 complex F2 (1H) points, 1024 complex F3 (1H) points, and six points of gradient strength on F1 were recorded from two scans per F1 and F2 increment. The spectral width obtained for F3 was 16 ppm. The maximum gradient strength for F2 was 1.2 G/cm and the minimum was -1.2 G/cm. The increments were at equal intervals. Six increments in the variable counter list used for F1 were 25, 50, 75, 100, 125, and 150 loop counters. Direct (F3) and indirect (F2) dimensions were zero-filled to 2048 and 128 points, respectively. Details of experimental parameters are shown in **Supplementary Tables 5 and 6**.”

Page 25

“**Fig. 4** An example application of T_2 -SMOOSY by solution-state NMR without MAS. **a** Fish feed pellets and D_2O were placed into 5-mm NMR tubes, and feed dissolution was evaluated. The black bars show the range of detected spatial z -position. **b–d** Pseudo-2D T_2 -SMOOSY spectral images. The T_2 -SMOOSY measurements were started approximately **b** 5 min, **c** 35 min, and **d** 65 min after sample preparation. The measurement times were approximately 5 min. The annotated major compounds are shown with relaxation times colored from red (short relaxation time) to blue (long relaxation time) according to the range shown in the bar above the figure. T_2 -weighted parameters were repetition time (TR) = 600 ms, TE = 1 ms, number of excitations (NEX) = 2, and matrix 1024×16 (chemical shift $\times z$ -position).”

The study should acknowledge a very recent report (in Analyst, DOI: 10.1039/d0an00118j) on HR-MAS CSI, which has ‘partially’ addressed both of these issues – sample integrity and metabolic displacement – with slow sample rotation (500 Hz) experiment, and with only one-hour acquisition on a non-chemically fixed intact organism.

[Response. 2-2]

The report suggested by the reviewers is important and interesting. Accordingly, we have added the text to include the citation.

Page 9 lines 14–20

“In addition, the tissue of the biological sample is immobilized, and the measurements will be different from those for metabolic profiling of living tissues. In a recent report on CSI using HR-MAS, a slow sample rotation (500 Hz) experiment

acquired metabolic profiling of a living organism in about an hour³⁷. Applying such pulse sequences and experimental approaches to SMOOSY may address concerns with the effects of MAS and are considered as areas for future study.”

37. Lucas-Torres, C. & Wong, A. Intact NMR spectroscopy: slow high-resolution magic angle spinning chemical shift imaging. *Analyst* **145**, 2520–2524 (2020). [10.1039/D0AN00118J](https://doi.org/10.1039/D0AN00118J)

Since spectral quality is a key component to metabolic profiling, the authors should consider showing the spectral quality of a 1D chemical shift spectrum (either with a standard 1D CPMG, or better yet, the individual slice F2 projection of HR-CSI of the different body component).

[Response. 2-3]

Spectral quality has been added to Supplementary Figure 2 and a citation to the References section.

Points-by-points response to Reviewer #3

The authors have made significant improvements to their manuscript. This greatly improved readability and I recommend publication. However, I do think the authors should consider including some of the more important figures from the supplementary material in the main text and possibly condense the descriptions of the experiments in the main text. Condensing the main text but including SI figures, while giving more detailed descriptions in the SI, will make the manuscript more accessible to a broader audience.

[Response. 3]

We have included an important figure related to the supplementary material in the main text as follows:

Page 25

Fig. 4 An example application of T_2 -SMOOSY by solution-state NMR without MAS. **a** Fish feed pellets and D_2O were placed into 5-mm NMR tube, and feed dissolution was evaluated. The black bars show the range of detected spatial z -position. **B–d** Pseudo-2D T_2 -SMOOSY spectral images. The T_2 -SMOOSY measurements were started approximately **b** 5 min, **c** 35 min, and **d** 65 min after sample preparation. The measurement times were approximately 5 min. The annotated major compounds are shown with relaxation times colored from red (short relaxation time) to blue (long relaxation time) according to the range shown in the bar above the figure. T_2 -weighted parameters were repetition time (TR) = 600 ms, TE = 1 ms, number of excitations (NEX) = 2, and matrix 1024×16 (chemical shift $\times z$ -position).”

We have also included a detailed description of the additional experiments as follows:

Page 4 lines 26–30

“As examples of possible applications, we described comprehensive analyses of an intact biological sample and two heterogeneous systems such as diffusion of feeding stimulants from aquaculture feeds and transport of small molecules via a membrane filter. We also present an example of spectrum analysis using the SMOOSY processor.”

Page 10 lines 25 to Page 11 line 17

“Stimulants in aquaculture feeds, such as betaine, organic acids, amines, amino acids, saccharides, and their mixtures, are important for the development of new aquaculture feeds^{38–40}. Therefore, the second example was fish feed diffusion (**Supplementary Figure 17b**). Feed pellets were placed at the bottom of an NMR tube, D_2O was added, and the diffusion of components of the feed was evaluated. Peptides have a low diffusion coefficient, e.g. ca. 6.81×10^{-11} (m²/s) and remain at the bottom of the NMR tube. Small molecules, such as amino acids, organic acids, and saccharides, have large diffusion coefficients; they dissolve and diffuse widely. Especially, the diffusion coefficients for lactate and acetate were large, ca. 1.74×10^{-9} (m²/s). Thus, the diffusion of several compounds into the feed was significant with rapid dissolution. To detect the dissolution of compounds into the feed, high-speed measurements immediately after sample preparation is necessary.

Thereafter, T_2 -SMOOSY was adjusted to about 5 min per measurement. The measurement was started at an early stage after sample preparation, and the dissolution process of the compounds into the feed was assessed (**Fig. 4**). The dissolution of lactate and acetate was confirmed with relaxation times of about 1.61 s (**Fig. 4b**). In addition, the dissolution of TMAO, betaine, taurine, choline, creatine, dimethylamine (DMA), and saccharides was confirmed, whereas the dissolution of amino acids was confirmed 35 min after sample preparation (**Fig. 4c**). A peptide signal was also detected. This finding might be explained by the prolonged relaxation time that was due to affinity with water or dissolution to improve the mobility. After 65 min (**Fig. 4d**), dissolution and affinity with water proceeded further. These compounds in solution are likely to attract fish. T_2 -SMOOSY over a short time frame may have problems with deterioration of spatial resolution and detection sensitivity; however, it is sufficient for the preliminary evaluation of samples. These results indicate that the technique is applicable to research on feed development including exploration of attractant compounds for fishes. Details of the above experiments are present in the **Supplementary Information**.”

”

38. Carlberg, H., Cheng, K., Lundh, T. & Brannas, E. Using self-selection to evaluate the acceptance of a new diet formulation by farmed fish. *Appl. Anim. Behav. Sci.* **171**, 226–232 (2015). [10.1016/j.applanim.2015.08.016](https://doi.org/10.1016/j.applanim.2015.08.016)

39. Hill, H. A., Trushenski, J. T. & Kohler, C. C. Utilization of soluble canola protein concentrate as an attractant enhances production performance of sunshine bass fed reduced fish meal, plant-based diets. *J. World Aquacult. Soc.* **44**, 124–132 (2013). [10.1111/jwas.12005](https://doi.org/10.1111/jwas.12005)

40. Kasumyan, A. O. & Doving, K. B. Taste preferences in fishes. *Fish Fish.* **4**, 289–347 (2003). [10.1046/j.1467-2979.2003.00121.x](https://doi.org/10.1046/j.1467-2979.2003.00121.x)

Page 12 lines 23–27

The experiment with fish feed shows that reducing the number of measurement points and scans, T_2 -SMOOSY can be measured in about 5 min. This time frame is sufficient for applications wherein data must be collected over a short period. However, such use is likely to involve a sacrifice of the spatial resolution and detection sensitivity.”

Page 13 lines 16–21

“For feed dissolution experiments, approximately 50 mg of Himezakura fish feed pellets (HIGASHIMARU Co., Ltd., Kagoshima, Japan) were placed into 5-mm NMR tubes after freeze-drying. Thereafter, 500 μ L of 0.1 M phosphate buffer solution [0.1 M K_2HPO_4/KH_2PO_4 (KPi); pH 7.0]/ D_2O buffer with 1 mM sodium 2,2-dimethyl-2-silapentane-5-sulfonate- d_6 (DSS- d_6) was added. T_2 -SMOOSY measurements began 5 min after sample preparation.”

Page 14 lines 16–29

“**Solution-state NMR experiments for a heterogeneous system.** The solution-state NMR spectra of a heterogeneous system of fish feed pellets in KPi/ D_2O were recorded using an Avance NEO-700 spectrometer (Bruker, Billerica, MA) equipped with an inverse triple-resonance cryogenic probe with a z -axis gradient for 5-mm diameter samples; this system was operated at 700.15 MHz for 1H and 176.06 MHz for ^{13}C . T_2 -SMOOSY spectra were recorded at 298 K. The analysis was performed to evaluate fish feed dissolution. For this, 16 complex F2 (1H) points, 1024 complex F3 (1H) points, and six points of gradient strength on F1 were recorded from two scans per F1 and F2 increment. The spectral width obtained for F3 was 16 ppm. The maximum gradient strength for F2 was 1.2 G/cm and the minimum was -1.2 G/cm. The increments were at equal intervals. Six increments in the variable counter list used for F1 were 25, 50, 75, 100, 125, and 150 loop counters. Direct (F3) and indirect (F2) dimensions were zero-filled to 2048 and 128 points, respectively. Details of experimental parameters are shown in **Supplementary Tables 5 and 6.**”

REVIEWERS' COMMENTS:

Reviewer #2 (Remarks to the Author):

The revised text has responded appropriately, and I have no more comments.